# Conformational dynamics between transmembrane domains and allosteric modulation of a metabotropic glutamate receptor

Vanessa A Gutzeit[1], Jordana Thibado[2], Daniel Starer Stor[2], Zhou Zhou[3], Scott C Blanchard[2,3,4], Olaf S Andersen[2,3], Joshua Levitz[2,4,5]*

[1]Neuroscience Graduate Program, Weill Cornell Graduate School of Medical Sciences, New York, United States; [2]Physiology, Biophysics and Systems Biology Graduate Program, Weill Cornell Graduate School of Medical Sciences, New York, United States; [3]Department of Physiology and Biophysics, Weill Cornell Medicine, New York, United States; [4]Tri-Institutional PhD Program in Chemical Biology, New York, United States; [5]Department of Biochemistry, Weill Cornell Medicine, New York, United States

**Abstract** Metabotropic glutamate receptors (mGluRs) are class C, synaptic G-protein-coupled receptors (GPCRs) that contain large extracellular ligand binding domains (LBDs) and form constitutive dimers. Despite the existence of a detailed picture of inter-LBD conformational dynamics and structural snapshots of both isolated domains and full-length receptors, it remains unclear how mGluR activation proceeds at the level of the transmembrane domains (TMDs) and how TMD-targeting allosteric drugs exert their effects. Here, we use time-resolved functional and conformational assays to dissect the mechanisms by which allosteric drugs activate and modulate mGluR2. Single-molecule subunit counting and inter-TMD fluorescence resonance energy transfer measurements in living cells reveal LBD-independent conformational rearrangements between TMD dimers during receptor modulation. Using these assays along with functional readouts, we uncover heterogeneity in the magnitude, direction, and the timing of the action of both positive and negative allosteric drugs. Together our experiments lead to a three-state model of TMD activation, which provides a framework for understanding how inter-subunit rearrangements drive class C GPCR activation.
DOI: https://doi.org/10.7554/eLife.45116.001

## Introduction

G-protein-coupled receptors form an extremely diverse family of membrane signaling proteins that play central roles in nearly all physiological processes and serve as the most frequent class of drug targets in biology (*Lagerström and Schiöth, 2008*). Metabotropic glutamate receptors (mGluRs) form a particularly important family of GPCRs in the brain, where they work in concert with iono-tropic iGluRs to control glutamatergic transmission (*Reiner and Levitz, 2018*). Based on their central roles in basic synaptic neurobiology along with compelling preclinical and clinical evidence, mGluRs serve as potential drug targets for a wide range of neurological and psychiatric diseases (*Nicoletti et al., 2011*). This physiological and clinical significance motivates studies on the mGluR activation process and the molecular mechanisms of mGluR-targeting drugs.

mGluRs, and other class C GPCRs, have a unique domain structure consisting of large, bi-lobed extracellular ligand binding domains (LBDs) and cysteine-rich domains (CRDs) that are coupled to

*For correspondence:
jtl2003@med.cornell.edu

seven-helix transmembrane domains (TMDs) which are structurally homologous among all GPCRs (*Niswender and Conn, 2010*). Furthermore, mGluRs constitutively dimerize in living cells (*Doumazane et al., 2011*; *Levitz et al., 2016*), and dimerization is required for glutamate-driven G-protein activation (*El Moustaine et al., 2012*). This unique domain structure and dimeric architecture raises many questions about the allosteric processes that underlie receptor activation. Recent single-molecule imaging studies have provided a dynamic interpretation of existing dimeric LBD crystal structures (*Kunishima et al., 2000*; *Muto et al., 2007*; *Tsuchiya et al., 2002*) to provide a model of the intrasubunit (i.e. LBD closure) and intersubunit (i.e. dimer interface reorientation) rearrangements that mediate the initial stages of mGluR activation (*Olofsson et al., 2014*) (*Levitz et al., 2016*; *Vafabakhsh et al., 2015*). However, despite major progress on the inter-subunit interactions and conformational dynamics of the extracellular LBDs, the processes that govern TMD dimer assembly and activation are comparatively less well understood. It remains unclear if TMDs can form stable interactions with each other, whether any such interactions are state-dependent and if potential inter-TMD rearrangements are driven autonomously or depend on allosteric input from the LBDs. A recent breakthrough cryo-electron microscopy study of full-length mGluR5 showed clear inter-TMD interactions in a glutamate-bound state (*Koehl et al., 2019*), further motivating investigation of inter-subunit coordination.

From a pharmacological perspective, understanding mGluR activation at the TMD level is crucial because allosteric modulators that bind within the TMD are utilized as both basic research tools and as potential clinical leads (*Foster and Conn, 2017*; *Lindsley et al., 2016*). Positive allosteric modulators (PAMs) are thought to primarily modulate agonist-induced activity, but have also been reported to directly elicit receptor activation in full-length receptors. It has also been established that isolated mGluR TMDs can initiate G protein signaling in responses to PAMs (*El Moustaine et al., 2012*; *Goudet et al., 2004*). Negative allosteric modulators (NAMs) typically inhibit agonist-driven activation, but their mechanism of action and effects on basal activity remain unclear. Unambiguous functional interpretation of mGluR-targeting allosteric drugs, with physiological readouts, is needed to further resolve their precise effects. Furthermore, a direct readout of the conformational impact of allosteric modulators at the TMD in the context of the plasma membrane of living cells is lacking, limiting the ability to characterize the relative affinity, efficacy, and kinetics of different compounds at the level of the receptor itself. Ultimately, a detailed understanding of PAMs and NAMs on receptor conformation, assembly and function is needed to use and develop drugs for both mechanistic studies and therapeutic applications.

Here, we use a battery of electrophysiological and imaging-based assays to show that positive allosteric modulators of mGluR2 serve directly as agonists which can drive activation by reorienting TMD dimers independently of allosteric input from the LBDs. Using a single-molecule subunit counting assay, we find that this inter-TMD reorientation is underscored by a unique, high propensity for mGluR2 TMD dimerization that is not seen in other group I and II mGluR subtypes or in canonical class A GPCRs. Using a new inter-TMD FRET assay we find that mGluR2 PAMs show variable apparent affinity, efficacy, kinetics, and reversibility of mGluR2 modulation which should inform future applications and drug development. We also find that NAMs modulate inter-TMD arrangement and can either be neutral antagonists or serve as inverse agonists. Our observations lead to a model of mGluR gating at the level of the TMD that accounts for the complex effects of allosteric drugs and motivates further work aimed at unraveling the heterogeneity of GPCR-targeting drugs.

## Results

### Positive allosteric modulators directly activate mGluR2 with minimal contribution from the extracellular domains

A comprehensive understanding of class C GPCR activation and signaling requires a mechanistic description of the effects of both LBD-targeting 'orthosteric' and TMD-targeting 'allosteric' compounds (*Figure 1A*). Classically, positive allosteric modulators (PAMs) of GPCRs have been defined by their ability to amplify the effects of orthosteric compounds without directly activating the receptor (*May et al., 2007*) (*Lindsley et al., 2016*; *Wacker et al., 2017*). However, some studies have shown both modulation and direct activation of GPCRs by PAMs (*May et al., 2007*), including mGluRs (*Rovira et al., 2015*) (*O'Brien et al., 2018*). Most previous studies, however, relied on end-

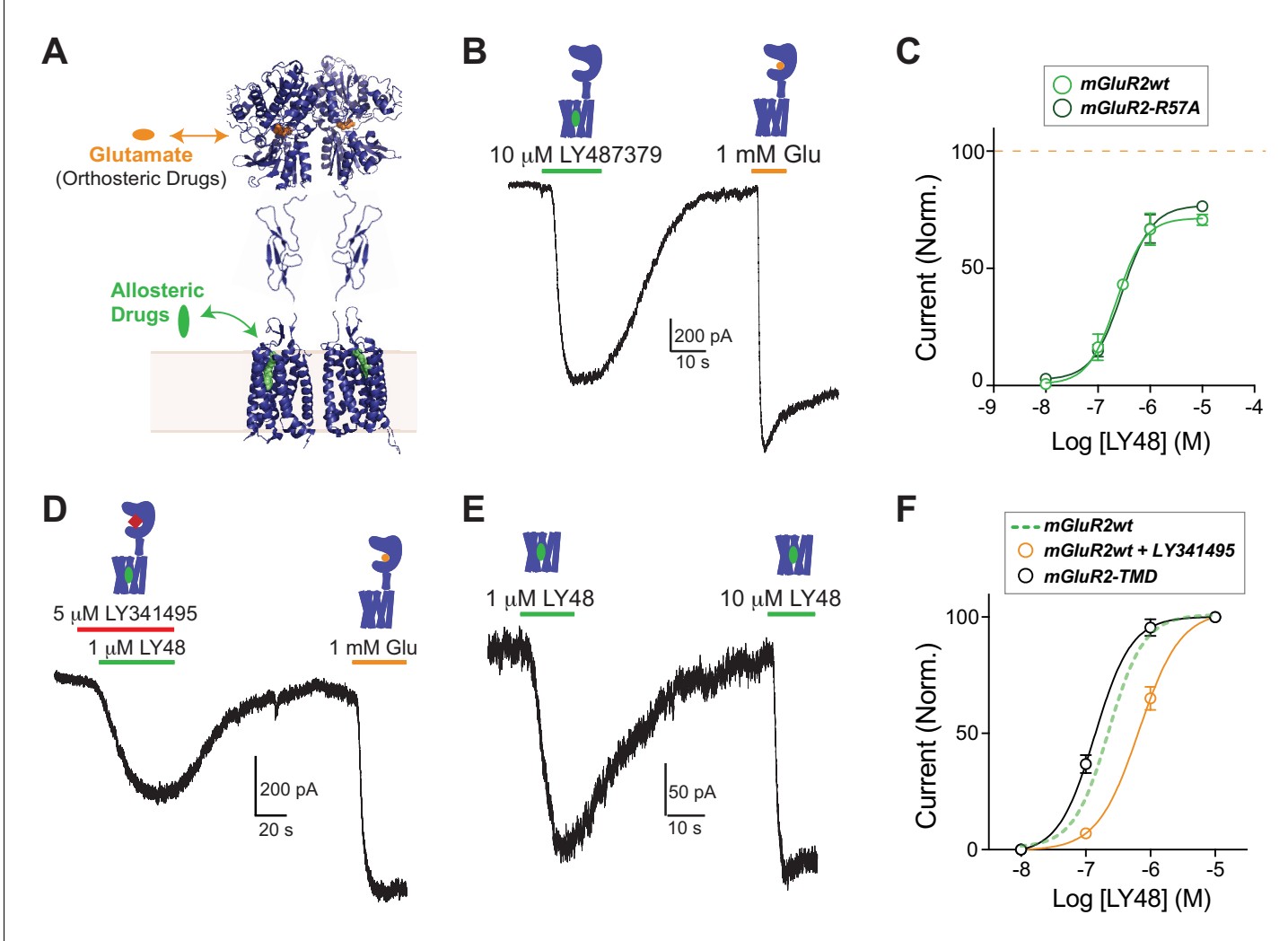

**Figure 1.** Positive allosteric modulators activate mGluR2 in a GIRK activation assay independently of ligand binding domains. (A) Structural model of a full-length mGluR based on structures of isolated domains (mGluR2 Ligand Binding Domain in Closed-Closed/Active state with two bound glutamate molecules = PDB 5CNI; mGluR2 Cysteine-Rich Domain = PDB 5KZQ; mGluR1 Transmembrane Domain = PDB 4OR2) showing the locations of the orthosteric and allosteric binding sites. (B) Representative whole cell patch clamp recording from HEK 293T cells expressing full-length mGluR2 showing an inward GIRK current induced by the positive allosteric modulator LY483739 (LY48) that is comparable to the response to saturating glutamate. (C) LY48 GIRK activation dose response curves for wild-type SNAP-mGluR2 (EC$_{50}$ = 0.23 ± 0.04 μM) and SNAP-mGluR2-R57A (EC$_{50}$ = 0.29 ± 0.08 μM). Values are normalized to saturating glutamate and come from at least three cells per conditions. Error bars show s.e.m. (D–E) LY48 GIRK responses are not blocked by a saturating concentration of the competitive orthosteric antagonist LY341495 (D) or by the removal of the extracellular domain of mGluR2 (E). (F) LY48 GIRK activation dose response curves showing the apparent affinity shifts observed when LY341495 is co-applied (orange; EC$_{50}$ = 0.66 ± 0.07 μM) or when the isolated mGluR2 TMD (black; EC$_{50}$ = 0.14 ± 0.01 μM) is tested. Values are normalized to saturating LY48 and come from at least three cells per conditions. Error bars show s.e.m. Note: N-terminally SNAP-tagged constructs were used for all recordings.

DOI: https://doi.org/10.7554/eLife.45116.002

The following figure supplements are available for figure 1:

**Figure supplement 1.** Further characterization of mGluR2 activation by LY483739.
DOI: https://doi.org/10.7554/eLife.45116.003
**Figure supplement 2.** Activation of mGluR2 by the mGluR2 PAMs TASP 043386 and CBiPES.
DOI: https://doi.org/10.7554/eLife.45116.004
**Figure supplement 3.** PAMs do not induce an inter-LBD FRET change in mGluR2.
DOI: https://doi.org/10.7554/eLife.45116.005

point assays and thereby failed to provide dynamic information about the onset and reversibility of PAM effects. This issue is exacerbated when studying mGluRs because mammalian cells secrete amino acids (*Tora et al., 2018*), obscuring whether PAMs can autonomously activate the receptor or merely modulate endogenous glutamate-driven activation.

To overcome these limitations, and to clearly detect PAM-induced effects on mGluR2 signaling with temporal precision, we used an electrophysiology-based assay of GPCR-mediated activation of G-protein-coupled inwardly-rectifying potassium (GIRK) channels in HEK 293T cells. In this system, activation of $G_{i/o}$-coupled receptors rapidly and reversibly produces potassium currents and constant perfusion of the bath prevents accumulation of endogenously released glutamate. GIRK channels are common, native effectors of group II/III mGluRs throughout the mammalian nervous system (*Dutar et al., 2000*) (*Watanabe and Nakanishi, 2003*) and provide a physiologically relevant read-out of receptor activation. We initially focused on the canonical mGluR2 PAM LY487379 (LY48) (*Johnson et al., 2003*). Application of only LY48 to cells expressing mGluR2 and GIRK produced large, reversible inward currents that were up to ~70% in amplitude compared to those induced by saturating glutamate (*Figure 1B*; *Figure 1—figure supplement 1A*). LY48 responses were blocked by co-application of the mGluR2 negative allosteric modulator (NAM) MNI 137 (*Figure 1—figure supplement 1B*). We also examined the LY48-induced effects on mGluR2 signaling using a calcium imaging assay, where a G-protein chimera (*Conklin et al., 1993*) permits a $G_{i/o}$-coupled receptor to signal via the Gq pathway to release calcium from intracellular stores. Consistent with the GIRK activation results, we saw clear agonism in response to LY48 application, which produced responses with a similar amplitude to glutamate (*Figure 1—figure supplement 1C,D*). To rule out any effects of ambient glutamate, we performed the GIRK measurements with mGluR2-R57A, a mutant with an ~30 fold reduction in glutamate affinity (*Malherbe et al., 2001*). mGluR2-R57A displayed very similar responses to LY48 (*Figure 1—figure supplement 1E*) and the LY48 dose–response curves for mGluR2-WT and mGluR2-R57A were identical with both showing maximal PAM-induced activation of ~70% relative to glutamate and an $EC_{50}$ of ~300 nM (*Figure 1C*). The direct agonist effect of LY48 on mGluR2 prevented precise measurement of potential modulatory effects on glutamate responses at LY48 concentrations $\geq$ 100 nM. 50 nM LY48, the highest concentration that did not produce a response on its own, had no effect on the glutamate-sensitivity of mGluR2 (*Figure 1—figure supplement 1F*), indicating that PAM activation and modulation likely occur over the same concentration range. Notably, two other PAMs, CBiPES and TASP 043386, also produced reversible, dose-dependent activation of mGluR2 (*Figure 1—figure supplement 2*).

To elucidate the role of the extracellular ligand binding domain (LBD) in PAM-driven agonism, we explored whether LBD closure, a key initial step in glutamate-mediated mGluR activation (*Kunishima et al., 2000*), is required for LY48 activation. Co-application of a saturating concentration of LY341495 (LY34), a competitive antagonist that prevents LBD closure (*Vafabakhsh et al., 2015*) did not prevent mGluR2 activation by LY48 (*Figure 1D*). Furthermore, consistent with previous studies (*El Moustaine et al., 2012*), complete removal of the extracellular domain did not prevent LY48-mediated activation, indicating that an intact LBD is not needed for mGluR2 activation (*Figure 1E*). We titrated LY48 in each construct and found subtle effects on LY48 agonism (*Figure 1F*). LY34 shifted the LY48 dose-response curve to the right, indicating that LBD closure weakly promotes PAM agonism. In contrast, removal of the extracellular domains shifted the dose–response curve to the left, consistent with a model where the LBD provides tonic inhibition of TMD activation by PAMs that is relieved either by LBD closure or by removal of the ECD. Further supporting this model, binding studies have shown up to a 10-fold leftward shift in PAM binding to mGluR2 in the presence of glutamate (*Doornbos et al., 2016*; *O'Brien et al., 2018*). Despite the modulatory effects of the LBD on PAM activation, saturating concentrations of LY48 were unable to directly induce LBD closure as assayed using an inter-LBD FRET assay (*Figure 1—figure supplement 3*).

Together these results show that PAMs are able to serve as direct allosteric agonists of mGluR2 and that this activation is only weakly modulated by the LBD. We next sought to further understand the intersubunit TMD interactions that drive activation of mGluRs.

## mGluR TMDs form dimers of variable propensity and assembly of mGluR2-TMD dimers is insensitive to allosteric drugs

Dimerization of class A GPCRs has been a controversial topic, although various experimental techniques suggest a model of transient or context-dependent dimer formation (*Gurevich and Gurevich,*

*2018*) (*Sleno and Hébert, 2018*). In contrast, class C GPCRs have long been known to form constitutive dimers (*Romano et al., 1996*). Most recently, we and others have shown that mGluRs form strict dimers in living cells and that this is driven primarily by interactions between the LBDs (*Doumazane et al., 2011*; *Levitz et al., 2016*). Furthermore, FRET-based studies show that inter-LBD reorientation is a key initial step in mGluR activation (*Doumazane et al., 2013*; *Olofsson et al., 2014*; *Vafabakhsh et al., 2015*), and that these LBD motions are tuned by inter-LBD interactions (*Levitz et al., 2016*). Much less is known about interactions between mGluR TMDs and whether a stable interface, or interfaces, is formed and how this might change during activation. A cross-linking study on full-length mGluR2 found evidence for constitutive interaction between TMDs (*Xue et al., 2015*) and the crystal structure of the mGluR1 TMD was solved as a cholesterol-mediated dimer (*Wu et al., 2014*). In contrast, a FRET study showed no evidence for dimerization between mGluR2 TMDs (*El Moustaine et al., 2012*) and the mGluR5 TMD was crystallized as a monomer (*Doré et al., 2014*). Most recently, cryo-electron microscopy structures of full-length mGluR5 showed an inter-TMD interface only in detergent micelles and in the presence of glutamate and a PAM, but not in lipid nanodiscs in the apo state (*Koehl et al., 2019*).

Given the weak, modulatory effects of the extracellular LBDs on mGluR2 activation by PAMs (*Figure 1D–F*) and the inconsistent information about inter-TMD interactions, we focused on isolated TMDs to probe the assembly and interactions between these domains using a single-molecule imaging-based approach termed 'SiMPull' (Single-Molecule Pulldown). This method allows detergent-solubilized receptor complexes to be immobilized from fresh cell lysate via antibodies on a glass coverslip to permit single-molecule imaging with TIRF microscopy (*Jain et al., 2011*). Photobleaching step analysis of individual complexes allows for precise determination of receptor stoichiometry and, unlike FRET or BRET-based methods, this assay is not sensitive to conformation or relative fluorophore orientation. We previously used SiMPull to show that mGluRs form strict dimers, and to map the major hotspots of inter-LBD interaction, which allowed us to conclude that inter-TMD interactions contribute to dimerization (*Levitz et al., 2016*). However, this work relied on GFP-tagged receptors which leads to a lysate that contains both surface-targeted and intracellular receptors, potentially including immature or partially degraded protein that can confound analysis.

To restrict our analysis to surface-targeted receptors, we expressed N-terminally SNAP-tagged receptor variants and labeled them with membrane-impermeable fluorophores. Expression and labeling of SNAP-tagged full-length mGluR2 or mGluR2-TMD in HEK 293T cells with the benzylguanine-conjugated fluorophore BG-LD555 (Materials and methods) showed surface labeling with minimal fluorescence inside the cell (*Figure 2A*; *Figure 2—figure supplement 1A*). Consistent with previous studies, following immobilization with a biotinylated anti-HA antibody, single SNAP-mGluR2 molecules photobleached primarily in one-step (~40%) or two-step (~60%) events with a small population showing ≥3 steps (5.5 ± 0.61%) (*Figure 2—figure supplement 1B,C*), consistent with an obligatory dimer with ~80% SNAP labeling efficiency (Materials and methods). We next performed the same measurements with SNAP-mGluR2-TMD and observed ~45% of spots bleaching in 2-steps (*Figure 2B–D*), consistent with a population of ~60% dimers. The SNAP-mGluR2-TMD protein displayed a similar proportion of ≥3 step events (4.9 ± 0.41%) as SNAP-mGluR2 (*Figure 2C*), indicating a lack of higher order complexes or non-specific aggregation of the isolated TMD. Importantly, the enhanced stability of the LD-555 fluorophore allowed us to improve signal-to-noise ratio and fluorophore lifetime, allowing for more accurate determination of photobleaching steps (*Figure 2—figure supplement 1D*). We next tested whether TMD dimerization is dependent on dilution of detergent, which in some cases induces assembly of GPCRs (*Jastrzebska et al., 2004*). Our initial measurements were done in 0.1% IGEPAL, which is above the critical micelle concentration (*Ghosh et al., 2004*) but substantially lower than the 1.2% used for cell lysis. SiMPull measurements of SNAP-mGluR2-TMD in 1.2% IGEPAL showed a very similar distribution of photobleaching steps with ~40% of molecules bleaching in two steps (*Figure 2—figure supplement 1E*). SiMPull experiments using *n*-dodecyl-*β*-D-maltopyranoside ('DDM'), a commonly used detergent that is compatible with purified mGluRs (*Wu et al., 2014*) (*Doré et al., 2014*), again observed ~40% two-step photobleaching but with a slightly higher proportion of larger aggregates (*Figure 2—figure supplement 1E*). Finally, to determine whether dimerization is influenced by protein expression level, we transfected cells with either a typical (0.7 μg) or very low (0.07 μg) amount of SNAP-mGluR2-TMD DNA; even when SNAP-mGluR2-TMD was expressed with only 0.07 μg of DNA, ~40% two-step photobleaching was still observed (*Figure 2—figure supplement 1F*).

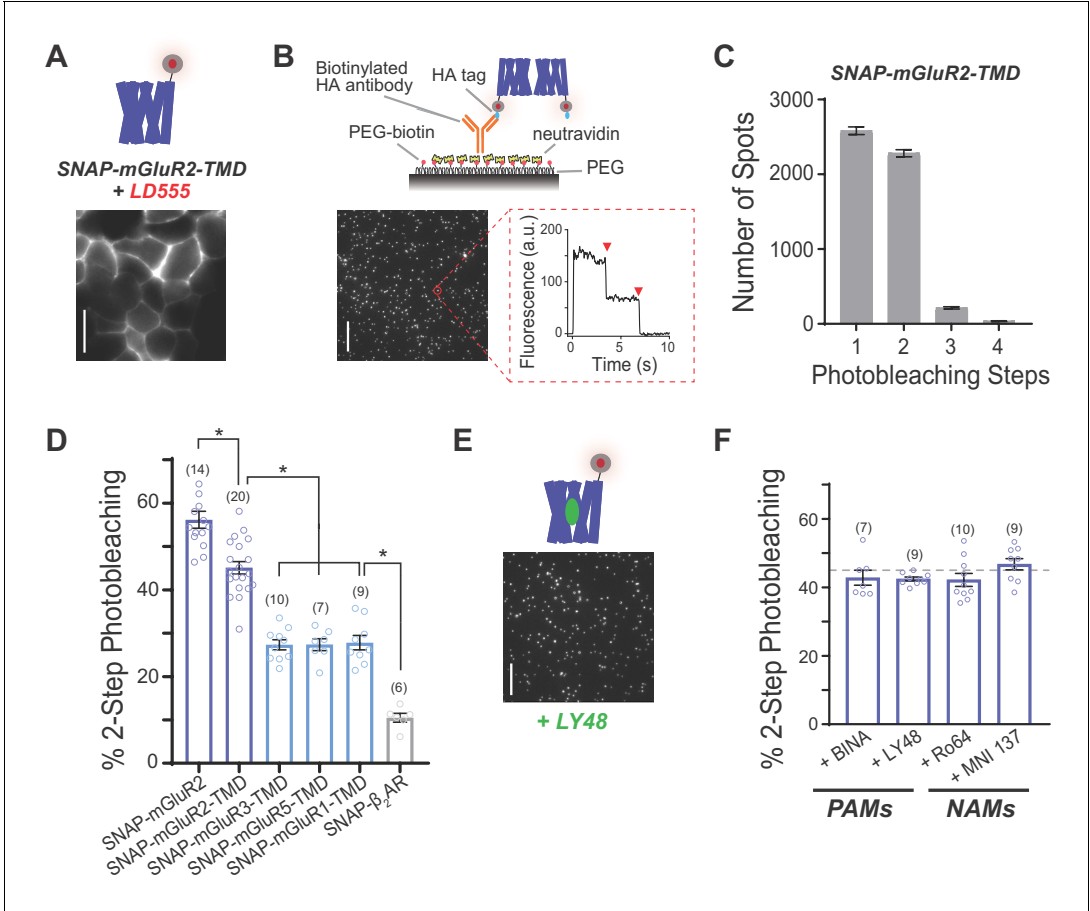

**Figure 2.** Single-molecule photobleaching analysis shows that mGluR transmembrane domains dimerize in the absence or presence of allosteric drugs. (A) Representative image showing expression and surface labeling of SNAP-mGluR2-TMD construct in HEK 293T cells prior to lysis. Scale bar = 10 μm (B) Single-molecule pulldown of SNAP-mGluR2-TMD reveals ECD-independent dimerization of surface-targeted TMDs. Top, schematic showing the SiMPull assay using a PEG-passivated glass coverslip and a biotinylated anti-HA antibody. Bottom, representative image of single molecules with representative fluorescence time course for an individual molecule (red circle) showing two-step photobleaching (red arrows). Scale bar = 10 μm. (C) Histogram summarizing the distribution of photobleaching step number for all SNAP-mGluR2-TMD molecules tested (n = 5113 spots from 20 movies). (D) Summary bar graph showing the percentage of spots bleaching in two steps for a range of N-terminally SNAP-tagged constructs labeled with LD555. * indicates statistical significance (unpaired t tests; for SNAP-mGluR2 vs. SNAP-mGluR2-TMD, p=0.0001; for SNAP-mGluR2-TMD vs. SNAP-mGluR3-TMD, SNAP-mGluR5-TMD or SNAP-mGluR1-TMD, p=3E-10, 3E-8, and 2E-7; for SNAP-ß2AR vs. SNAP-mGluR3-TMD, SNAP-mGluR5-TMD or SNAP-mGluR1-TMD, p=4E-8, 1E-6, and 1E-6. The number of movies analyzed for each condition is shown in parentheses. Error bars show s.e.m. (E) Representative image showing immobilized SNAP-mGluR2-TMD molecules treated with LY48. Scale bar = 10 μm. (F) Summary bar graph showing the lack of an effect of application of PAMs (10 μM LY48 or 1 μM BINA) or NAMs (10 μM Ro 64–5229 or 1 μM MNI 137) on SNAP-mGluR2-TMD stoichiometry. Dotted gray line shows the % two-step bleaching observed for un-liganded SNAP-mGluR2-TMD. The difference between conditions, including WT, were not significant (ANOVA, p=0.2). The number of movies analyzed for each condition is shown in parentheses. Error bars show s.e.m.

DOI: https://doi.org/10.7554/eLife.45116.006

The following figure supplements are available for figure 2:

**Figure supplement 1.** Further analysis of single-molecule pulldown of mGluR2 and mGluR2-TMD.

DOI: https://doi.org/10.7554/eLife.45116.007

**Figure supplement 2.** Comparative analysis of GPCR TMD stoichiometry: mGluR2, mGluR3, mGluR5 and ß2AR Representative HEK 293T expression.

DOI: https://doi.org/10.7554/eLife.45116.008

**Figure supplement 3.** Further analysis of the lack of effect of ligands on mGluR2 stoichiometry.

DOI: https://doi.org/10.7554/eLife.45116.009

We next wondered if a propensity for inter-TMD dimerization is unique to mGluR2 or if it could be observed for other mGluRs. We tested mGluR3, the other group II mGluR, and both mGluR1 and mGluR5, the group I mGluRs. Consistent with previous studies with GFP-tagged receptors (*Levitz et al., 2016*), SNAP-mGluR3 and SNAP-mGluR5 showed photobleaching step

distributions consistent with dimerization (48.9 ± 1.3% for mGluR3 and 48.4 ± 1.4% for mGluR5). Compared to SNAP-mGluR2-TMD, SNAP-mGluR1-TMD, SNAP-mGluR3-TMD and SNAP-mGluR5-TMD all exhibited a lower two-step photobleaching frequency (~30%) (*Figure 2D*; *Figure 2—figure supplement 2*), which suggests that the strength and extent of inter-TMD interaction is mGluR subtype-specific.

We also tested if a canonical class A GPCR would show a similar behavior under the same conditions. We focused on the beta-2 adrenergic receptor (ß₂AR), which has been observed to dimerize or oligomerize under some conditions (*Angers et al., 2000*; *Hebert et al., 1996*) (*Fung et al., 2009*; *Jain et al., 2011*; *Mathiasen et al., 2014*), but is functional as a monomer (*Whorton et al., 2007*) and has been crystallized extensively as a monomer (*Rasmussen et al., 2011*) (*Masureel et al., 2018*). We expressed HA- and SNAP-tagged ß₂AR ('SNAP-ß₂AR') and pulled down surface receptors from HEK 293T cell lysate via the same anti-HA antibody and under identical lysis conditions to the mGluRs (*Figure 2—figure supplement 2*). We observed ~10% two-step photobleaching which is consistent with a pure monomer (Materials and methods) (*Figure 2D*). Importantly, expression was similar to mGluR2-TMD for all other GPCRs tested (*Figure 2—figure supplement 2*). In addition, we recently reported that GFP-tagged opsin also exists primarily as a monomer when expressed in HEK 293T cells and immobilized under similar conditions (*Pandey et al., 2017*). Together this indicates that mGluR TMDs show a unique, high propensity for dimerization that is not seen across all GPCR subtypes.

Given the prominent, but incomplete, dimerization of SNAP-mGluR2-TMD, we wondered if altering TMD conformation with allosteric drugs would modify receptor stoichiometry. To test this, we incubated SNAP-mGluR2-TMD expressing cells and lysates with PAMs or NAMs during the SiMPull experiment. Previous work has shown that mGluR2 PAMs can bind and exert conformational effects on mGluR2 under identical detergent conditions (*Vafabakhsh et al., 2015*). *Figure 2E* shows isolated SNAP-mGluR2-TMD molecules in the presence of a saturating concentration of LY48 which did not produce a visible change in spot size or promote aggregate formation. We counted SNAP-mGluR2-TMD bleaching steps in the presence of either two different mGluR2 PAMs (LY48 and BINA) or two different mGluR2 NAMs (MNI 137 and Ro 64–2259) and found that the photobleaching step distributions in all cases were similar, with ~40–45% two-step bleaching events (*Figure 2E,F*; *Figure 2—figure supplement 3A*) and no change in the proportion of higher order complexes with >3 bleaching steps (<10%). Additionally, we tested if the addition of glutamate or a PAM would alter the ~60% two-step bleaching events observed for SNAP-mGluR2, as a recent study suggested that higher order mGluR2 complexes can form under some conditions (*Møller et al., 2018*), and found that two-step bleaching events remained at ~60%, whether or not saturating concentrations of glutamate or glutamate plus LY48 were added to cells and lysate during SiMPull (*Figure 2—figure supplement 3B*).

These results show that mGluR2 TMDs readily dimerize with an enhanced propensity compared to other mGluRs and canonical class A GPCRs. Given that the dimerization propensity of mGluR2 TMDs is not altered by ligand binding, we reasoned that TMD rearrangement, rather than alterations in assembly, must occur upon mGluR2 activation by either orthosteric or allosteric drugs.

## An inter-TMD FRET assay reveals LBD-independent, activation-associated intersubunit rearrangement

Our results, along with the work of others, has established that mGluR2 activation can be initiated at either the LBDs or the TMDs (*Figure 1*) and that both LBDs and TMDs form dimeric interfaces (*Figure 2*). Whereas mutagenesis and intersubunit FRET studies have shown that inter-LBD rearrangement is an important early step in receptor activation (*Levitz et al., 2016*), less is known about inter-TMD rearrangement. Crucially, it remains unclear if inter-TMD motion is driven solely by inter-LBD conformational changes, which brings the lower lobes of the LBD closer together to produce a more extensive interface, or if TMD rearrangement is able to occur autonomously without the LBDs. In previous studies, fluorescent proteins were inserted into the intracellular loops or C-termini of mGluRs and glutamate-induced FRET changes, which were highly dependent on the precise insertion site for their directionality, were measured (*Marcaggi et al., 2009*; *Tateyama et al., 2004*; *Tateyama and Kubo, 2006*; *Yanagawa et al., 2011*) (*Grushevskyi et al., 2019*; *Hlavackova et al., 2012*). Taken together these studies suggest that inter-TMD reorientation is part of the activation process. Unfortunately, in all cases, G-protein activation was prevented by the modification to the

construct, challenging the interpretation of the study since the receptor was likely unable to reach a fully active conformation. Furthermore, all previous studies focused on either orthosteric ligands or LBD-targeting trivalent ions without assessing the effects of allosteric drug binding to the TMD.

We therefore sought to develop a FRET sensor that reports on activation-associated motions driven by allosteric drugs and hypothesized that an N-terminal fluorophore on the isolated TMD should report on any inter-TMD rearrangement while allowing the receptor to maintain G-protein coupling (*Figure 3A*). To test this, we co-expressed SNAP-mGluR2-TMD and CLIP-mGluR2-TMD in HEK 293T cells and labeled the cells with donor (CLIP-Surface 547) and acceptor (BG-LD655) fluorophores for CLIP and SNAP, respectively. The donor fluorophore was excited with a 561 nm laser and both donor and acceptor channels were imaged simultaneously using a dual camera imaging system (*Figure 3A*; *Figure 3—figure supplement 1A*; see Materials and methods). As a control, we bleached the acceptor fluorophore and observed recovery of the donor fluorescence, confirming that substantial FRET can occur (*Figure 3—figure supplement 1B*). Consistent with the enhanced dimerization of the mGluR2-TMD, a smaller donor recovery was observed with the ß$_2$AR (*Figure 3—figure supplement 1B*). Strikingly, application of LY48 produced a readily detectable FRET increase, as determined by a simultaneous increase in acceptor fluorescence and decrease in donor fluorescence, which was reversed with drug washout (*Figure 3B*; *Figure 3—figure supplement 1C*). LY48-induced FRET responses were repeatable (*Figure 3C*), showed no desensitization over 5 min (*Figure 3—figure supplement 1D*), were blocked by the NAM MNI 137 (*Figure 3D*) and dose-dependent over the concentration range used for functional experiments (*Figure 3E,F*). Indistinguishable results were observed whether we co-expressed SNAP- and CLIP-tagged mGluR2-TMD or just SNAP-mGluR2-TMD and labeled with a mixture of BG-conjugated donor and acceptor dyes (*Figure 3—figure supplement 1E*). Importantly, SNAP-tagged mGluR2 TMDs showed LY48-induced GIRK activation (*Figure 1E*), indicating that they remain capable of activating G proteins, as expected. To test if LBD-independent inter-TMD rearrangement is a feature of other mGluRs, we asked if donor and acceptor conjugation to SNAP-mGluR5-TMD would allow for the detection a FRET response to a PAM. Application of the mGluR5 PAM VU0360172 produced a reversible FRET response suggesting that, indeed, this conformational assay is generalizable across the mGluR family (*Figure 3—figure supplement 1F*).

To further probe the connection between the conformational changes detected in this assay and receptor activation, we tested the effects of G-protein coupling on inter-TMD FRET changes. A classical model of GPCRs is the formation of a ternary complex with a G-protein heterotrimer that enhances the receptor's affinity for agonists (*Stadel et al., 1980*). Consistent with this model, when we co-expressed a dominant-negative Gα protein with reduced guanine nucleotide affinity to stabilize the receptor–G protein interaction (*Barren and Artemyev, 2007*), there was a substantial left-shift in the inter-TMD FRET dose–response curve (*Figure 3F*; *Figure 3—figure supplement 2*). A similar left-shift in inter-LBD FRET in the presence of this G-protein mutant was reported previously (*Doumazane et al., 2013*). Together these experiments show that allosteric activation of mGluR2 leads to inter-TMD rearrangement and that the conformational change can be driven autonomously by the TMDs without initiation by the LBDs.

The large FRET changes upon application of a PAM, with a similar concentration-dependence as the downstream activation, motivated us to probe the relationship between the inter-TMD FRET signal and the structural rearrangements that occur upon activation. Structural and spectroscopic studies, primarily on class A GPCRs, have led to a general model of 7-TM domain activation involving inter-helix interactions that are modified by ligand binding and converge on an outward motion of TM6 which opens up a G protein binding site (*Weis and Kobilka, 2018*) (*Thal et al., 2018*). A number of microswitches involving helices 3, 5, and 6 have been proposed to mediate the conformational rearrangements that drive receptor activation. How these microswitches contribute to PAM agonism in mGluRs is not well understood, especially in the context of intersubunit reorientation. Asp735 in helix 5 has been proposed as a central residue within a 'trigger switch' (*Figure 3—figure supplement 3A*) that mediates PAM binding-induced conformational changes for most, but not all, mGluR2 PAMs (*O'Brien et al., 2018*; *Pérez-Benito et al., 2017*; *Schaffhauser et al., 2003*). Consistent with our prediction, mutation of asparagine 735 to aspartate (N735D) resulted in drastically reduced activation by LY48 in the GIRK assay (*Figure 3—figure supplement 3B,C*). We next asked if our inter-TMD FRET response is also dependent on interactions involving N735. Consistent with functional measurements, the N735D mutant showed a ~ 20 fold rightward shift in the LY48 inter-

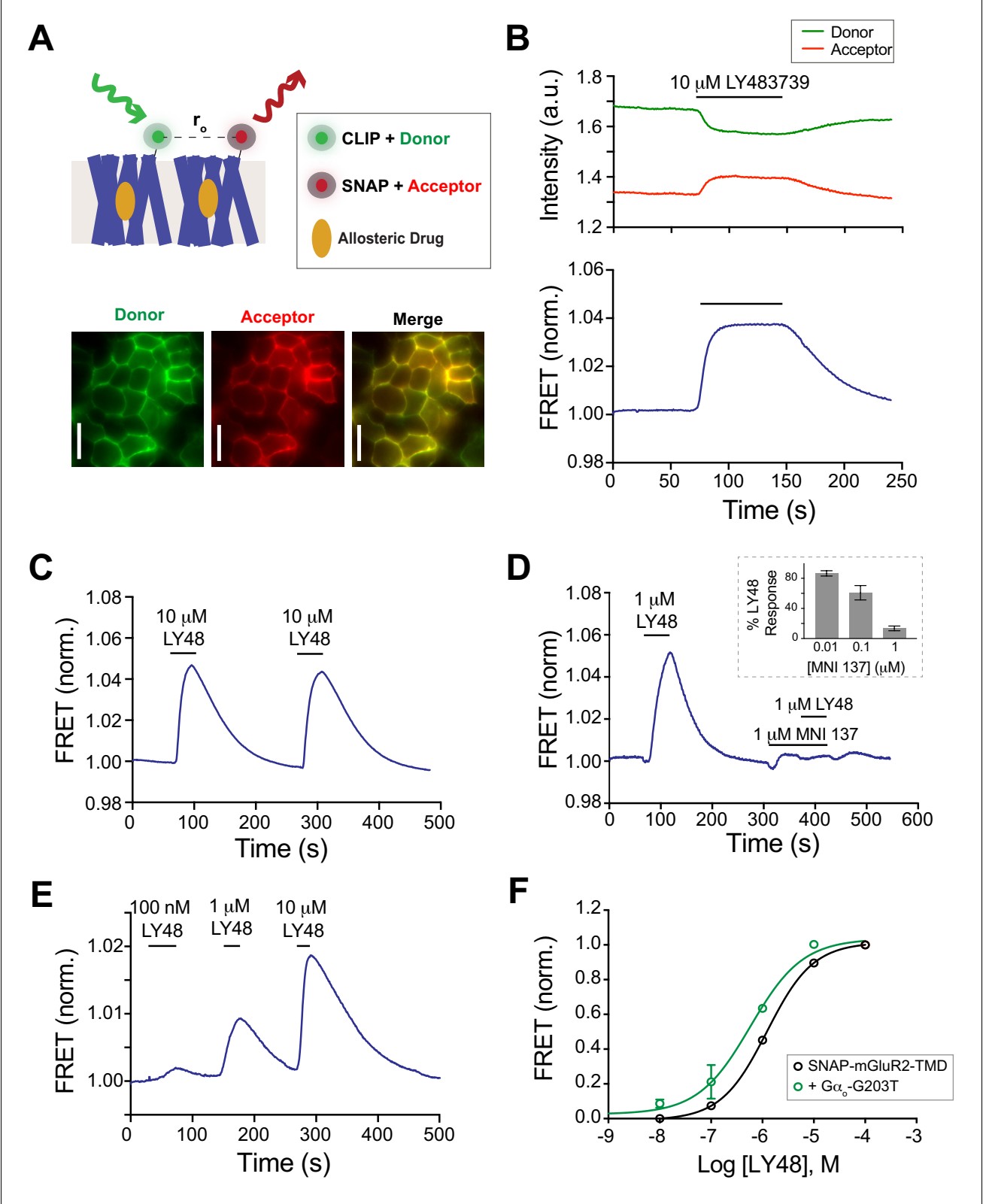

**Figure 3.** An inter-TMD FRET assay reveals LBD-independent reorientation in response to positive allosteric modulators. (**A**) Top, schematic showing SNAP- and CLIP-tagged mGluR2-TMD constructs labeled with donor and acceptor fluorophores. Bottom, images showing donor and acceptor channels following donor excitation with a 561 nm laser. Scale bars = 40 μm. (**B**) Representative time course showing donor and acceptor fluorescence intensity (top) during LY48 application. Baseline-normalized FRET is shown in the bottom trace, revealing a large, reversible increase in response to

*Figure 3 continued on next page*

*Figure 3 continued*

LY48 application. (C–E) Representative traces showing that LY48-induced inter-TMD FRET increase is repeatable (**C**), blocked by the NAM MNI 137 (**D**), and dose-dependent (**E**). The inset to (**D**) shows the extent of block of a 10 µM LY48 response by different concentrations of MNI 137. (**F**) Dose–response curve for LY48-induced FRET increase for SNAP-mGluR2-TMD ($EC_{50}$ = 1.2 ± 0.1 µM), WT-SNAP-mGluR2 + dominant negative G protein ($EC_{50}$ = 0.6 ± 0.04 µM). The dose–response curves were significantly different (two-way ANOVA, p=0.002). Values are normalized to saturating LY48 and come from at least three separate experiments per conditions. Error bars show s.e.m.

DOI: https://doi.org/10.7554/eLife.45116.010

The following figure supplements are available for figure 3:

**Figure supplement 1.** Further characterization of an inter-TMD mGluR FRET Assay.

DOI: https://doi.org/10.7554/eLife.45116.011

**Figure supplement 2.** Further analysis of G protein effects on inter-TMD rearrangement.

DOI: https://doi.org/10.7554/eLife.45116.012

**Figure supplement 3.** A role for a TMD 'Trigger Switch' in allosteric agonism and inter-TMD re-arrangement.

DOI: https://doi.org/10.7554/eLife.45116.013

TMD FRET dose–response curve compared to the wild-type receptor (*Figure 3—figure supplement 3D,E*). These results show that inter-TMD FRET changes and PAM agonism depend on TMD micro-switches associated with intrasubunit conformational changes.

## PAM-induced inter-TMD FRET responses are correlated with functional PAM affinity and efficacy

An increasingly important aspect of GPCR pharmacology is the ability of different drugs, that act via the same general binding site, to induce different signaling effects. For instance, agonists may be biased to different pathways, show drastically different efficacies or display different kinetics (*Wacker et al., 2017*). To explore this question in mGluRs, we investigated a panel of commercially available and widely used mGluR2 PAMs (*Figure 4A*) to test if all compounds show a similar FRET response and if our assay can reveal differences that are masked by less sensitive read-outs. Similar to LY48, all the PAMs tested produced a clear, dose-dependent increase in inter-TMD FRET (*Figure 4B*; *Figure 4—figure supplement 1A,B*). *Figure 4C* shows dose–response curves for all four PAMs tested in the inter-TMD FRET assay. At saturating concentrations, however, the PAMs produced different maximal FRET changes, as compared to saturating LY48 (*Figure 4C*; *Figure 4—figure supplement 1C*). TASP 0433864 produced a maximum FRET increase of ~1.3 times that of LY48, whereas BINA produced an increase only ~0.8 times that of LY48. Importantly, there was no correlation between the apparent PAM affinity and the apparent efficacy, as measured by maximal TMD FRET change (*Figure 4D*).

Given the sensitivity of both our functional GIRK and our conformational inter-TMD FRET assays, we tested a recently reported potential endogenous PAM of mGluR2. Xanthurenic acid (XA), a metabolite in the kynurenic acid pathway, has been reported to be an allosteric activator of mGluR2 with nanomolar potency (*Copeland et al., 2013*; *Fazio et al., 2016*). This observation has been controversial (*Fazio et al., 2017*), motivating further analysis with more direct readouts. We applied XA at concentrations up to 50 µM and observed neither inter-TMD FRET changes (*Figure 4—figure supplement 2A*) nor GIRK currents in full-length mGluR2-expressing cells (*Figure 4—figure supplement 2B*). These experiments highlight the potential of the inter-TMD FRET assay for drug screening and show that XA does not serve as an allosteric agonist of mGluR2; it likely exerts its effects indirectly on mGluR2 signaling in physiological settings.

We next sought to further probe the variation in PAM-induced inter-TMD FRET changes. While *Figure 1* shows that mGluR2 PAMs by themselves serve as potent agonists, they are also known to enhance the response to orthosteric agonists. We used an inter-LBD FRET assay to assess the relative strength of allosteric modulation by different mGluR2 PAMs. Agonists induce a decrease in FRET between fluorophore-conjugated N-terminal SNAP- and CLIP-tags (*Figure 4E*). Previous studies showed that LY48 primarily enhances the maximal FRET change induced by a partial agonist, such as DCG-IV, rather than the apparent affinity for the agonist (*Doumazane et al., 2013*). We tested this effect for all PAMs by applying a saturating concentration of DCG-IV followed by co-application of a saturating concentration of each PAM, which produced further FRET decreases (*Figure 4F,G*). Similar to LY48, all the tested PAMs modulated the DCG-IV response, but did not

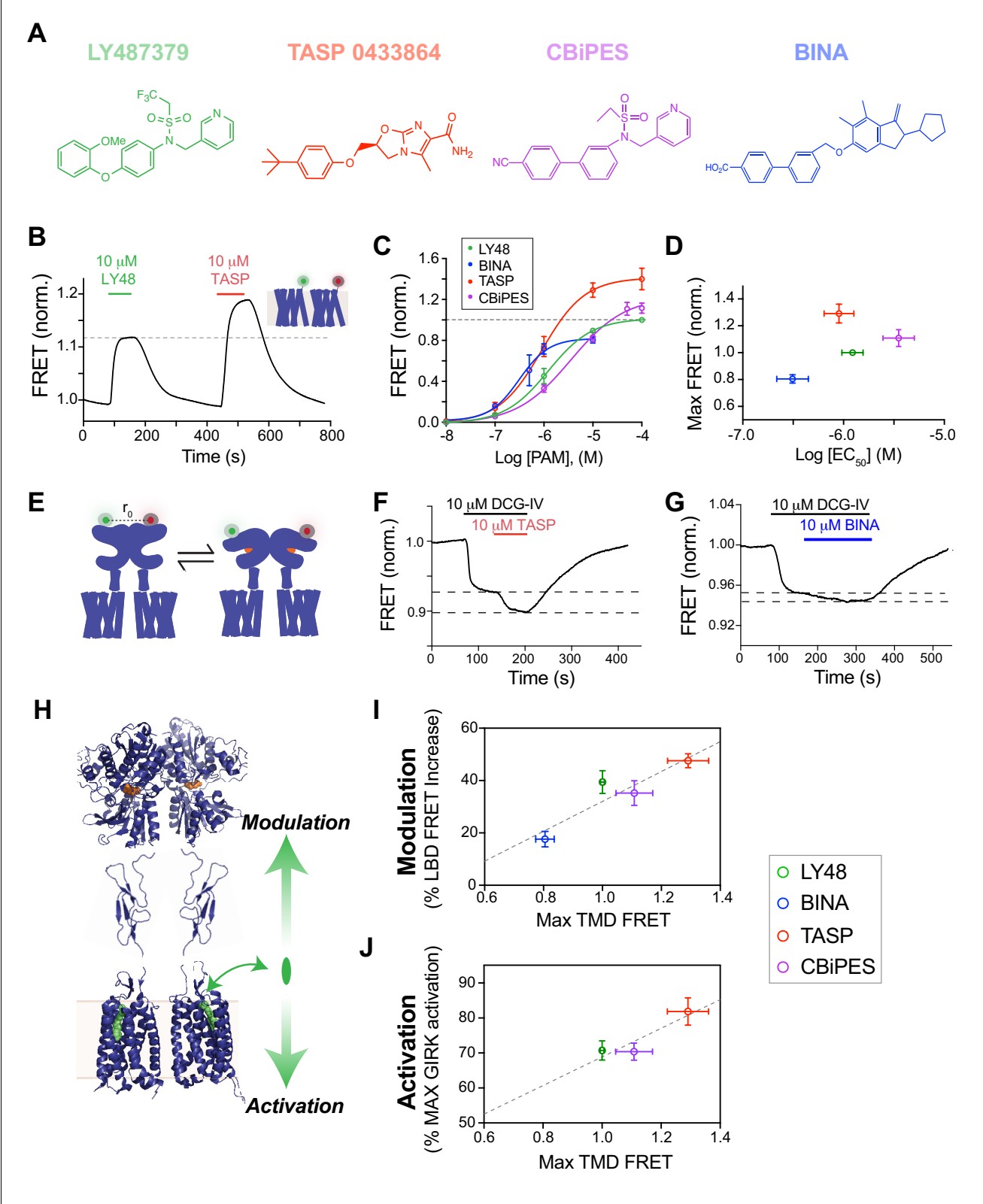

**Figure 4.** Structurally distinct PAMs increase inter-TMD FRET with variable potency and efficacy which is correlated to the strength of both allosteric activation and modulation. (**A**) Chemical structures of mGluR2 PAMs. (**B**) Representative trace showing inter-TMD FRET responses to LY48 and TASP. At saturating concentrations, TASP shows a larger FRET response than LY48. (**C**) Dose-response results for all four PAMs tested in the inter-TMD FRET assay, normalized to saturating LY48. (LY48 $EC_{50}$ = 1.2 ± 0.1 μM, TASP $EC_{50}$ = 0.9 ± 0.1 μM, CBiPES $EC_{50}$ = 3.5 ± 0.2 μM, BINA $EC_{50}$ = 3.1 ± 0.2 μM).

*Figure 4 continued on next page*

*Figure 4 continued*

Values are normalized to saturating LY48 and come from at least three separate experiments per condition. Error bars show s.e.m. (D) Summary of inter-TMD efficacy (i.e. max amplitude) and apparent affinity (i.e. $EC_{50}$) for all PAMs tested. Note the lack of correlation between relative efficacy and relative affinity. (E) Schematic of inter-LBD FRET assay where SNAP-tagged full-length mGluR2 constructs are labeled with donor and acceptor fluorophores and an orthosteric agonist is applied, leading to a decrease in FRET. (F–G) Representative inter-LBD FRET traces showing variability of the magnitude of PAM potentiation of the response to a saturating concentration of an mGluR2 agonist (DCG-IV). TASP (F) produces a larger increase in the FRET response than BINA (G). (H) Schematic showing that PAMs are able to exert their functional effects on mGluR2 in two different directions, either through alteration of the response to orthosteric agonists ('modulation') or through direct control of receptor signaling ('activation'). Error bars show s.e.m. (I–J) Plots showing correlation between inter-TMD FRET efficacy and either allosteric modulation (I; $R^2$ = 0.84) or allosteric activation (J; $R^2$ = 0.84).

DOI: https://doi.org/10.7554/eLife.45116.014

The following figure supplements are available for figure 4:

**Figure supplement 1.** Further comparative analysis of PAM-mediated inter-TMD FRET changes and GIRK activation.

DOI: https://doi.org/10.7554/eLife.45116.015

**Figure supplement 2.** Xanthurenic acid does not serve as an allosteric agonist of mGluR2 representative inter-TMD FRET.

DOI: https://doi.org/10.7554/eLife.45116.016

**Figure supplement 3.** Comparison of allosteric activation and modulation efficacies of mGluR2 PAMs.

DOI: https://doi.org/10.7554/eLife.45116.017

alter LBD FRET in the absence of an orthosteric agonist (*Figure 4—figure supplement 3A*). This result demonstrates that mGluR2 PAMs are able to both modulate the response to agonists, as assessed with inter-LBD FRET measurements, and directly activate the receptor, as assessed with GIRK current measurements (*Figure 4H*). Strikingly, the maximal inter-TMD FRET change produced by a given PAM is correlated with its ability to both modulate the agonist response (*Figure 4I*; *Figure 4—figure supplement 3B*) and to directly produce G-protein activation (*Figure 4J*; *Figure 4—figure supplement 3C*).

Together these measurements reveal variations among PAMs with regard to both affinity and efficacy. This is reminiscent of what is observed with orthosteric agonists of mGluR2, which display a wide range of affinities and efficacies that are accurately reported by inter-LBD FRET measurements (*Doumazane et al., 2013*; *Vafabakhsh et al., 2015*). Importantly, in the case of PAMs both the efficacy of allosteric agonism and the modulation of orthosteric agonism are correlated with inter-TMD FRET changes, suggesting that the state stabilized by PAMs may be the same conformation stabilized by orthosteric agonists.

## Different PAMs produce different kinetics of TMD FRET changes and receptor activation

Given the temporal sensitivity and direct nature of our inter-TMD FRET assay, we decided to characterize the kinetics of FRET changes associated with our panel of PAMs. We first compared the timing of LY48-induced inter-TMD FRET changes with the inter-LBD FRET changes induced by glutamate (*Figure 5A*). Across all concentrations, both the induction and the reversal of inter-LBD FRET in response to glutamate was faster than the PAM-induced changes (*Figure 5B,C*). Based on the expected wash-in and wash-out times of our system (*Figure 5—figure supplement 1A,B*; see Materials and methods), we believe that glutamate-induced FRET changes are limited only by the time course of drug exchange, but the LY48 responses are slower than expected based on a simple model of binding and unbinding and show weak dose-dependence. We found similar slow kinetics for LY48-induced FRET changes in the inter-LBD FRET assay (*Figure 5—figure supplement 1*). These results suggest either that the accessibilities of orthosteric and allosteric binding sites or that the kinetics of the associated conformational changes differ, meaning that the kinetics of the downstream effects induced by agonists versus PAMs would differ. Indeed, saturating LY48-induced GIRK currents were significantly slower than saturating glutamate-induced GIRK currents (10 μM LY48 (time 10% to 90%)=14.3 ± 3.5 s versus 1 mM Glutamate (Time 10% to 90%)=1.9 ± 0.5 s; paired T-test, p=0.04).

We next compared the inter-TMD FRET kinetics for all four PAMs. To our surprise, while saturating concentrations of LY48, CBiPES, and TASP gave rise to clear FRET responses with relatively rapid onset (10–40 s) and reversal (<2 min), BINA showed drastically slower FRET responses that took up

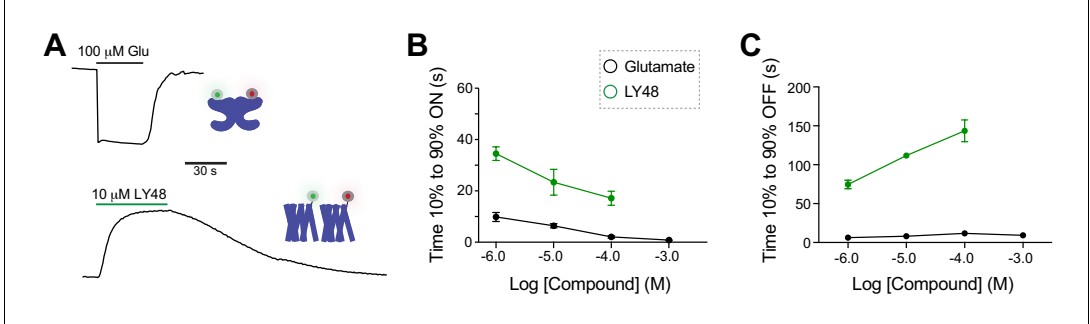

**Figure 5.** Comparative analysis of kinetics of glutamate-induced LBD FRET changes and LY48-induced TMD FRET changes. (**A**) Representative inter-LBD (top) and inter-TMD (bottom) FRET traces showing the timing of responses to saturating glutamate and LY48, respectively. (**B**) Summary graph showing the dose-dependence of the ON kinetics of ligand-induced inter-TMD or inter-LBD FRET changes. * indicates statistical significance between glutamate and LY48 responses at a given concentration (unpaired t tests: for 1 μM, p=0.02; for 10 μM, p=0.04; for 100 μM, p=0.03). (**C**) Summary graph showing the dose-dependence of the OFF kinetics of ligand-induced inter-TMD or inter-LBD FRET changes. * indicates statistical significance between glutamate and LY48 responses at a given concentration (unpaired t tests: for 1 μM p=0.0003; for 10 μM, p=0.0007 μM; for 100 μM p=0.001). Error bars show s.e.m. Values come from at least three separate measurements per conditions.

DOI: https://doi.org/10.7554/eLife.45116.018

The following figure supplement is available for figure 5:

**Figure supplement 1.** Further analysis of glutamate-induced inter-LBD kinetics and LY48-induced inter-TMD kinetics.
DOI: https://doi.org/10.7554/eLife.45116.019

to 3 min to reach a steady state and were irreversible on the time scale of our measurements for up to 10 min (**Figure 6A,B**). These distinct kinetics of PAM responses were maintained in our GIRK activation assay where LY48 showed fully-reversible currents, whereas BINA responses were maintained following up to 10 min of washout (**Figure 6C**). This irreversibility precluded obtaining reliable dose–response data for BINA in our GIRK activation assay. Furthermore, CBiPES also showed substantially faster ON and OFF kinetics relative to LY48 (**Figure 6A,B**; **Figure 6—figure supplement 1**). Interestingly, co-expression of dominant-negative G protein altered inter-TMD FRET kinetics, with accelerated ON and OFF LY48 responses (**Figure 6—figure supplement 2A**). This is consistent with G-protein-mediated stabilization of an active conformation, and suggests that the G-protein-stabilized active state facilitates PAM unbinding or exit from the core of the TMD.

We next asked why BINA-mediated responses are irreversible and reasoned that the most likely explanations are either that BINA simply binds and unbinds extremely slowly to the allosteric site within the mGluR2 TMD or that BINA, which is very hydrophobic (cLogP ≈ 7.8 versus 2.8-3.0 for LY48, CBi and TASP based on values in ChemAxon and ChemSpider databases), partitions into the plasma and organellar membranes which protects it from washout. To test this, we applied BINA followed by MNI 137, a NAM that blocks the LY48 FRET (**Figure 3D**) and GIRK activation responses (**Figure 1—figure supplement 1B**). MNI 137 clearly reversed the BINA-induced FRET increase back to baseline, implying that it indeed can rapidly compete with BINA for a common binding site (**Figure 6D**). However, following washout of MNI 137 a slow FRET increase was observed even though BINA was not reapplied. The results suggest that either there is a reservoir of BINA, and the FRET increase results from BINA rebinding from this reservoir, or that BINA and MNI bind at different sites and that MNI can reverse the BINA-induced conformational change even when BINA is bound to the receptor. To further probe this, we used a fluorescence quench assay for drug-induced changes of lipid bilayer properties (**Ingólfsson and Andersen, 2010**) (**Figure 6—figure supplement 3A**; see Materials and methods), in which changes in gramicidin channel function is used as a reporter of membrane modification. Of the four PAMs tested, BINA produced the greatest changes in gramicidin channel function (i.e. membrane properties) over a range of concentrations from 1 to 10 μM (**Figure 6—figure supplement 3B,C**). The persistent BINA response is thus consistent with a model where BINA readily partitions into the plasma membrane lipid bilayer (and possibly other lipidic compartments). This membrane binding would both limit BINA's availability to the receptor (slow the BINA-induced FRET increase) and serve as a reservoir that could greatly slow the washout

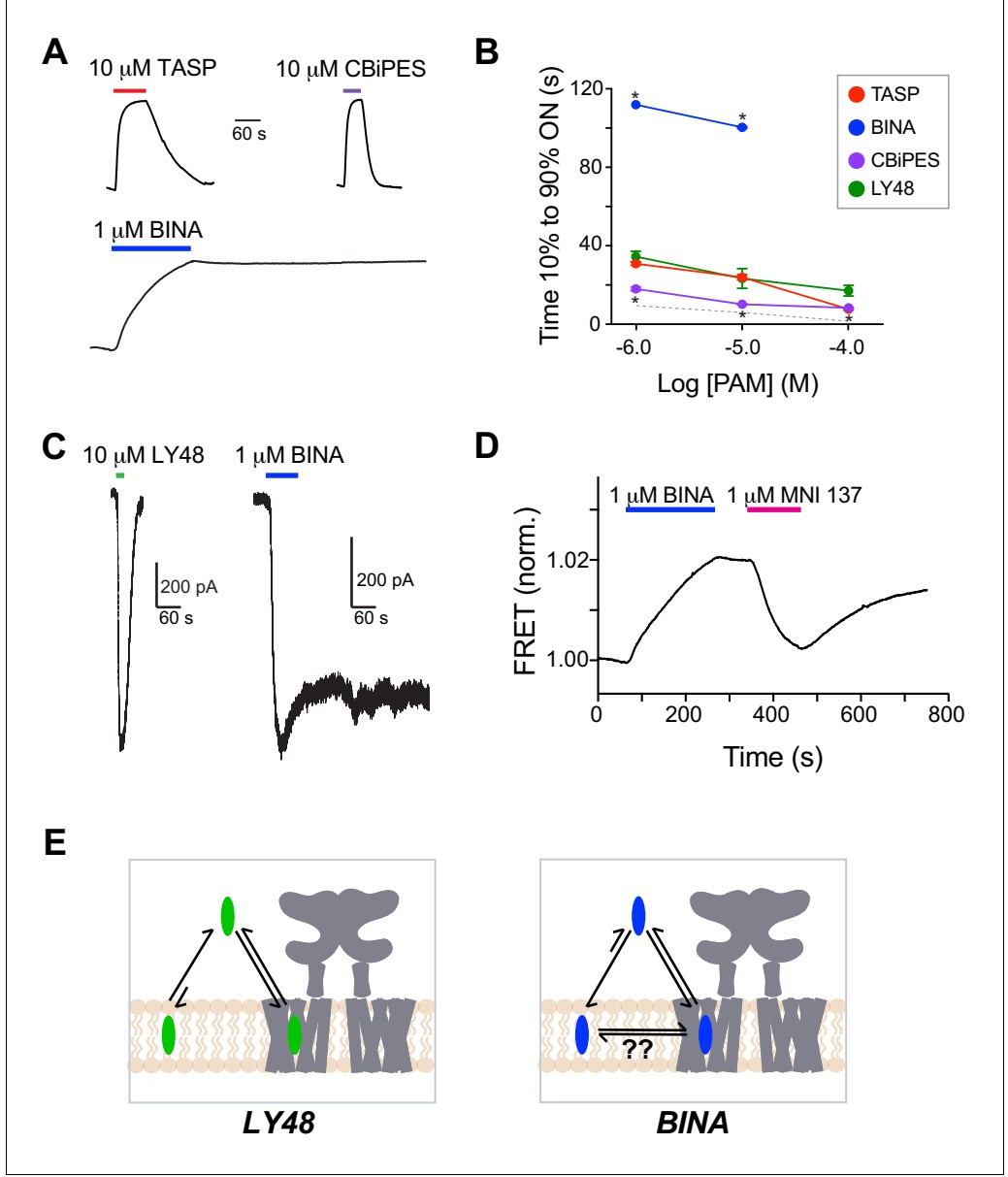

**Figure 6.** Different mGluR2 PAMs alter inter-TMD FRET and receptor activation with distinct kinetics and reversibility. (**A**) Representative inter-TMD FRET traces for TASP, CBiPES and BINA showing slower onset and lack of reversibility for BINA. (**B**) Summary of dose-dependent ON kinetics for all four mGluR2 PAMs tested BINA was significantly slower (unpaired T-test; 1 μM: p=0.000002, 10 μM: p=0.007) and CBiPES was significantly faster (unpaired T-test; 1 μM: p=0.04, 10 μM: p=0.03, 100 μM: p=0.02) than LY48. At least three separate measurements were made for each condition. Error bars show s.e.m. The dotted gray line shows the values obtained for glutamate in the inter-LBD FRET response. (**C**) Whole cell patch clamp recordings showing reversibility of LY48 agonism and irreversibility of BINA agonism of full-length mGluR2. (**D**) Representative inter-TMD FRET trace showing the effect of application of an mGluR2 NAM (MNI 137) following BINA application. Notably, MNI 137 reverses the BINA-induced FRET increase but following MNI 137 washout, the FRET level increases again. (**E**) Schematics showing working model of PAM binding and membrane interaction for LY48 versus BINA. Unlike other PAMs, BINA may partition directly into the lipid bilayer from where it can access the allosteric binding site on mGluR2 either directly within the bilayer or following exit on the extracellular side.

DOI: https://doi.org/10.7554/eLife.45116.020

The following figure supplements are available for figure 6:

**Figure supplement 1.** OFF kinetics of PAM-induced conformational changes of mGluR2 .

DOI: https://doi.org/10.7554/eLife.45116.021

*Figure 6 continued*
**Figure supplement 2.** Analysis of the effects of G protein on LY48-induced inter-TMD FRET kinetics.
DOI: https://doi.org/10.7554/eLife.45116.022
**Figure supplement 3.** Analysis of the effects of G protein on LY48-induced inter-TMD FRET kinetics.
DOI: https://doi.org/10.7554/eLife.45116.023

of BINA from the system (*Figure 6E*). Our results do not resolve whether BINA is able to bind directly to the TMD via the plasma membrane, as has been proposed for some class A GPCR ligands (*Bokoch et al., 2018*; *Piechnick et al., 2012*), or if it needs to first exit the bilayer and then bind via the extracellular face of the TMD.

## Inter-TMD FRET reveals that mGluR2 NAMs serve as either neutral antagonists or inverse agonists which stabilize a high FRET state

Because analysis of PAMs using inter-TMD FRET changes revealed striking differences between compounds, we decided to also test commercially available NAMs (*Figure 7A*). Application of MNI 137 alone produced minimal FRET changes even at the highest concentrations tested (*Figure 6B*), suggesting that the resting TMD conformation is not altered, though our sensors may not detect very localized conformational changes. In contrast, Ro 64–5229 ('Ro 64') produced robust, dose-dependent inter-TMD FRET increases with amplitudes comparable to mGluR2 PAMs (*Figure 7B,C*; *Figure 7—figure supplement 1A*). Interestingly, Ro 64 showed similar ON kinetics to LY48 (*Figure 7— figure supplement 1B*), but very slow OFF kinetics that did not fully reverse on the time course of our experiments (*Figure 7B*; *Figure 7—figure supplement 1A*). However, unlike BINA, neither MNI 137 nor Ro 64 produced changes in gramicidin channel function indicating that they do not alter membrane properties (*Figure 7—figure supplement 1C*), despite their hydrophobicity (cLogP $\approx$ 1.7 for MNI 137 and $\approx$ 4.9 for Ro 64). These results suggest that although both MNI 137 and Ro 64 serve as allosteric antagonists, their underlying mechanisms differ.

To further investigate the distinct effects of MNI 137 and Ro 64, we tested the ability of both NAMs to inhibit the inter-LBD FRET response to an orthosteric agonist. Saturating concentrations of either MNI 137 or Ro 64 showed a similar level of partial antagonism of the FRET response to DCG-IV (*Figure 7—figure supplement 1D–F*). These results indicate that, while both NAMs can fully block the functional orthosteric agonist response, they produce only a partial effect on LBD conformation and the extent of this effect is similar for both NAMs despite different effects at the level of the TMD.

Based on the strong effect of Ro 64 on inter-TMD FRET, we hypothesized that it may alter receptor function in the absence of glutamate. To test this, we applied both NAMs to HEK 293T cells expressing GIRK and full-length mGluR2. Whereas MNI 137 showed no effect on basal current (*Figure 7D*), Ro 64 produced a small but clear outward current (*Figure 7E,F*). These data show that Ro 64 acts as an inverse agonist while MNI 137 is a neutral antagonist. Despite the effect on basal GIRK activity, Ro 64 did not alter the basal inter-LBD FRET, indicating that it exerts its effects without altering LBD conformation (*Figure 7—figure supplement 1G*), as is also seen with MNI 137 (*Figure 7—figure supplement 1H*). Consistent with Ro 64 serving as an inverse agonist and MNI 137 serving as a neutral antagonist, MNI 137 was able to partially reverse the inter-TMD FRET increase induced by Ro 64 (*Figure 7—figure supplement 1I*). Together these results suggest a three-state model of activation where PAMs and inverse agonists stabilize different high FRET TMD arrangements and neutral NAMs merely inhibit the effect of other allosteric drugs (*Figure 7G*).

## Discussion

### The role of inter-TMD interaction in mGluR modulation and activation

Together this work shows that the TMDs of mGluRs are able to directly interact with each other (*Figure 2*) and that this interaction is altered during the activation process (*Figures 3–7*), even in the absence of the extracellular domains. Based on the previously established constitutive dimerization of mGluRs (*Doumazane et al., 2011*; *Levitz et al., 2016*) and the inter-TMD affinity demonstrated here with SiMPull and subunit counting, it is reasonable to assume that the high local concentration

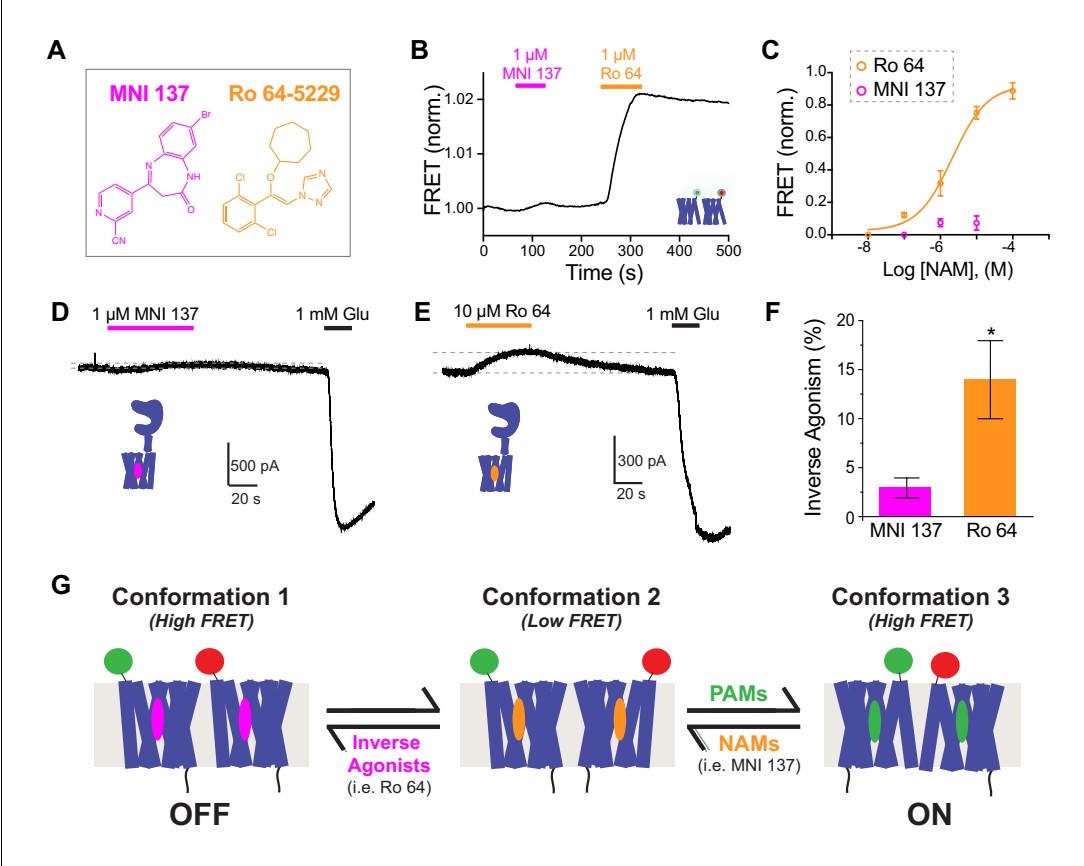

**Figure 7.** Inverse agonism or neutral antagonism of NAMs is associated with the presence or absence of an inter-TMD FRET Response. (**A**) Chemical structures of the mGluR2 NAMs, MNI 137 and Ro 64–5229. (**B**) Representative inter-TMD FRET trace showing the lack of a response for MNI 137, but a large FRET increase in response to Ro 64 application. (**C**) Inter-TMD FRET dose–response curves showing a large dose-dependent response for Ro 64 with an $EC_{50}$ of 2.1 ± 0.2 µM. Values are normalized to saturating LY48 and come from at least three separate experiments per conditions. Error bars show s.e.m. (**D–F**) Functional evidence for neutral antagonism of MNI 137 and inverse agonism of Ro 64. Representative current traces show no effect on baseline levels for MNI 137 (**G**), but a clear outward current for Ro 64 (**H**). Results are summarized in (**F**). Error bars show s.e.m. * indicates statistical significant (unpaired t-test, p=0.04). (**J**) Working three-state model accommodating a FRET increase for application of either PAMs or inverse agonists. A combination of distinct inter-TMD interfaces and differences in intra-TMD conformation likely underlie the different FRET values.
DOI: https://doi.org/10.7554/eLife.45116.024

The following figure supplement is available for figure 7:

**Figure supplement 1.** Further characterization of functional and conformational effects of mGluR2 NAMs.
DOI: https://doi.org/10.7554/eLife.45116.025

of TMDs in the context of full-length receptors ensures that inter-TMD interactions contribute to receptor assembly and activation. Indeed, an inter-TMD interaction between full-length mGluR5 subunits was seen in a recent cryo-electron microscopy study (*Koehl et al., 2019*). Although all mGluR TMDs tested in SiMPull showed a clear dimerization propensity, compared to the TMDs of mGluR1, mGluR3 and mGluR5 the TMD of mGluR2 showed a higher proportion of dimeric molecules. This suggests that the inter-TMD affinity is higher for mGluR2 and that there is heterogeneity within the mGluR family regarding inter-TMD interaction strength. Heterogeneity of this kind could indicate that there are intrinsic differences in the intersubunit interactions and related conformational changes that occur during the activation process between subtypes. Consistent with our observation of weaker inter-TMD interaction for mGluR5, Koehl et al did not observe a clear interface between TMDs in full-length mGluR5 when the receptor was imaged in the apo state in nanodiscs. Furthermore, the inability of prototypical class A GPCRs to form dimers in our assay suggests that the inter-TMD interactions of mGluRs are uniquely stable.

To decipher the conformational changes that occur at the level of TMD dimers, we report a new sensor for measuring inter-TMD FRET changes in response to allosteric drugs, which allowed us to detect a FRET change in response to PAM and NAM binding. Because inter-TMD dimerization of mGluR2 in SiMPull was not altered in the presence or absence of allosteric drugs, we interpret the PAM-induced inter-TMD FRET increase to report reorientation of existing TMD dimers rather than ligand-induced dimer formation. Furthermore, the ability of PAMs to serve as agonists in both full-length receptors and isolated TMDs (*Figure 1*), indicates that these conformational changes may represent a related activation pathway to that initiated by glutamate.

All the PAMs tested produced a FRET increase, suggesting a common conformational change, but the NAM Ro 64 also produced a FRET increase. Since the underlying conformations must be different between PAM and NAM-bound states and both states produce an increase relative to baseline, the simplest model to describe this data requires three states (*Figure 7G*). In this model, there is a resting state that produces low levels of basal activity, which can be stabilized by classical, neutral NAMs such as MNI 137. From this resting state, PAMs can rearrange the TMDs to produce a high FRET state and allosteric inverse agonists can also rearrange the TMDs to produce a different high FRET state. While caution is required when interpreting these results structurally, an attractive interpretation is that each state corresponds to a distinct inter-TMD interface. *Xue et al. (2015)* used inter-subunit crosslinking in full-length mGluR2 to propose a dimer reorientation model where a TM4/5 interface at rest rotates to form a TM6/TM6 interface upon activation. Similarly, *Koehl et al. (2019)* found that cross-linking between residues in TM6 led to constitutive activation of mGluR5. Such a rearrangement would indeed decrease the distance between the extracellular end of TM1 between subunits, potentially accounting for the FRET increase upon PAM application that we observe. Furthermore, the crystal structure of the mGluR1 TMD with a NAM shows a dimer mediated primarily by TM1 (*Wu et al., 2014*). Such a state could explain the FRET increase seen upon inverse agonist application in our assay. While further structural and functional work is needed to test this model, similar 'rolling' dimer interface models have recently been proposed for dimeric GPCR systems, including CXCR5 (*Jin et al., 2018*) and the neurotensin receptor (*Dijkman et al., 2018*).

It's worth mentioning that a recent study used fluorescence number and brightness analysis to infer the presence of ligand-dependent higher order oligomers of full-length mGluR2 in cultured neurons (*Møller et al., 2018*). While the formation of larger complexes could explain the ligand-induced FRET increases observed in our assay, this is contradicted by the observation that neither addition of PAM to mGluR2-TMDs or glutamate or glutamate and PAM to full-length mGluR2 led to changes in receptor stoichiometry in SiMPull. These results indicate that ligand-induced oligomer formation would not produce an interface as stable as the core dimeric interface.

Although there is clear evidence for inter-subunit rearrangement, it is important to note that intra-subunit conformational changes likely also contribute to changes in the FRET signal. *Hlavackova et al. (2012)* and *Grushevskyi et al., 2019* used fluorescent proteins inserted into intracellular loops of mGluR1 to find that FRET changes observed with inter-TMD sensors occur faster than those observed with intra-TMD sensors, suggesting a complex interplay between dimer rearrangement and TMD activation, where dimer reorientation may precede receptor activation. Consistent with this view, we find that the mutation N735D, which is known to hinder the intrasubunit triggering of TMD activation (*Schaffhauser et al., 2003*) (*Pérez-Benito et al., 2017*), right-shifted the inter-TMD FRET dose–response curve by a factor of 20 (*Figure 3—figure supplement 3D,E*). Future work is needed to thoroughly address the relationship between intrasubunit and intersubunit conformational changes and to determine if TMDs exert cooperative, functional effects on each other.

## Conformational and functional diversity of mGluR allosteric modulators

Our functional experiments demonstrate that PAMs serve directly as mGluR2 agonists, removing ambiguity about the functional effects of these compounds. Using a GIRK readout allowed us to test a biologically relevant effector of mGluR2 activation and observe the onset and offset of functional effects. Reports of allosteric agonist activity also exist for mGluR4 (*Rovira et al., 2015*), mGluR5 (*Rook et al., 2013*) and mGluR7 (*Mitsukawa et al., 2005*) indicating that agonist activity via the TMD binding site is likely a common feature of all mGluRs. In addition, we uncovered basal activity of mGluR2 that is driven at the level of the TMD which can be unmasked by inverse agonists, as was

seen with Ro 64. This extends our previous work that has shown, in contrast, minimal LBD-induced basal activity of mGluR2 (*Vafabakhsh et al., 2015*).

A key finding is that allosteric drug effects on mGluRs can occur independent of glutamate binding to the LBD, raising questions about the coupling mechanism between LBDs and TMDs. The inability of PAMs or NAMs to directly drive conformational changes at the LBD, the ability of PAMs to induce activation without concomitant LBD activation and, most importantly, the ability of isolated TMDs to rearrange autonomously together indicates that coupling between the LBDs and the TMDs is likely to be weak. Weak coupling, potentially in the form of flexible linkers between CRD and TMDs or flexibility within the CRDs, would, in principle, allow for significant rotation between TMDs as we and others propose. Importantly, there is evidence for various forms of coupling between LBDs and TMDs, and agonism at the LBD level is able to drive TMD activation. We show that LBD conformation modulates the apparent PAM affinity (*Figure 1F*) and that PAMs can modulate the efficacy of partial agonist-induced conformational changes at the LBD (*Figure 4E–G*). These observations suggest a model where LBDs provide tonic inhibition of the TMDs, which can either be relieved by agonist binding or overcome by PAM binding. Finally, the close correlation between efficacy of PAM agonism and PAM modulation indicates that both PAMs and orthosteric agonists ultimately drive the same conformational state at the level of the TMDs.

Finally, the temporal precision and sensitivity of our inter-TMD FRET assay allowed us to uncover previously unappreciated diversity in mGluR2-targeting allosteric compounds. PAMs display a range of efficacies as determined by both the maximal FRET change and the maximal activation of GIRK channels. These differences in efficacy presumably reflects variable abilities to stabilize an active conformation as has been observed in single-molecule FRET studies of LBD-targeting mGluR2 agonists (*Vafabakhsh et al., 2015*) and TMD-targeting $\beta_2$AR agonists (*Gregorio et al., 2017*). Detailed structural information is needed to understand the molecular basis for differences between PAMs, which likely originates with the precise pose of each compound within the TMD binding site.

The FRET assay allows us to distinguish subtle differences between PAMs and NAMs, which allowed us to demonstrate the lack of effect of Xanthurenic acid, ruling out the possibility that it serves as an endogenous mGluR2 allosteric modulator. More generally, the sensitivity of the method suggests that it has potential in drug screening.

Our FRET assay revealed major differences in kinetics between different PAMs. CBiPES had faster ON and OFF kinetics compared to LY48, whereas TASP showed similar kinetics to LY48, indicating that variability between ligands is also seen with kinetics and is not correlated with relative efficacy or affinity. Importantly, all PAM responses were slower ON and OFF than those observed with glutamate. Formation of a ternary complex with G proteins accelerated both the ON and OFF kinetics of LY48, which is the opposite to what has been reported in the $\beta_2$AR where G-protein coupling drastically slows agonist release through the engagement of a lid structure at the extracellular face of the receptor (*DeVree et al., 2016*).

Most strikingly, BINA showed extremely slow ON kinetics and complete irreversibility which we attribute, at least partially, to partitioning within the plasma and organellar membranes. Given the relative hydrophobicity of all mGluR PAMs, such a partitioning model could also explain the weak dose-dependence of OFF kinetics observed for LY48, TASP and CBiPES. Analogously, slow wash-out from lipidic compartments has previously been shown to produce dose-dependent OFF kinetics for hydrophobic drugs targeting GABA$_A$ receptors (*Gingrich et al., 2009*). Together these results show the potential importance of membrane effects on the action of GPCR-targeting drugs, especially hydrophobic TMD-targeting molecules. Finally, in contrast to BINA, the NAM Ro 64 showed fast ON and slow OFF kinetics despite the lack of membrane perturbation in an in vitro liposome assay. It's worth noting that while one would expect that very hydrophobic molecules, such as Ro 64, would be potent bilayer modifiers that is not always the case (*Alejo et al., 2013*; *Dockendorff et al., 2018*).

Recent work has shown that the kinetics of GPCR-targeting drugs are important determinants of their cellular and physiological effects (*Lane et al., 2017*). In the case of D2 dopamine receptors, for example, variable dissociation rates are major determinants of biased agonism (*Klein Herenbrink et al., 2016*), whereas the association rates of D2 receptor-targeting antipsychotics strongly predicted their ability to induce extrapyramidal side effects (*Sykes et al., 2017*). The conformational and kinetic variability of allosteric drugs are particularly important when assessing the in vivo effects of different drugs. Comparative in vivo studies of mGluR2 allosteric modulators have not been performed but our study argues that attention is needed to the relative affinity, efficacy and timing of

different PAMs and NAMs, especially in the preclinical context where drugs with precisely tuned pharmacological properties hold promise for psychiatric disease treatment (*Doumazane et al., 2011*).

# Materials and methods

**Key resources table**

| Reagent type | Designation | Source | Identifiers | Additional information |
|---|---|---|---|---|
| Cell line (*H. sapiens*) | HEK 293T | ATCC | ATCC Cat# CRL-3216, RRID:CVCL_0063 | |
| Transfected construct (*Rattus norvegicus*) | SNAP-mGluR2 | *Doumazane et al., 2011* | | |
| Transfected construct (*Rattus norvegicus*) | SNAP-mGluR2-TMD | *Doumazane et al., 2011* (modified) | | Following SNAP tag, mGluR2 truncatd to position Q558. |
| Transfected construct (*Rattus norvegicus*) | SNAP-mGluR3-TMD | *Doumazane et al., 2011* (modified) | | Following SNAP tag, mGluR3 truncatd to position E567. |
| Transfected construct (*H. sapiens*) | SNAP-mGluR5-TMD | *Doumazane et al., 2011* (modified) | | Following SNAP tag, mGluR5 truncatd to position V570. |
| Transfected construct (*Rattus norvegicus*) | SNAP-mGluR1-TMD | *Doumazane et al., 2011* (modified) | | Following SNAP tag, mGluR1 truncatd to position V583. |
| Transfected construct (*Rattus norvegicus*) | SNAP-mGluR2-R57A | *Doumazane et al., 2011* (modified) | | R at position 57 replaced with A. |
| Transfected construct (*Rattus norvegicus*) | EGFP-TM-GαoA-G203T | *Lober et al., 2006* (modified) | | G at position 203 replaced with T. |
| Transfected construct (*Rattus norvegicus*) | CLIP-mGluR2-TMD | *Doumazane et al., 2011* (modified) | | Following CLIP tag, mGluR2 truncatd to position Q558. |
| Transfected construct (*Rattus norvegicus*) | CLIP-mGluR2 | *Doumazane et al., 2011* | | |
| Transfected construct (*Rattus norvegicus*) | SNAP-mGluR2-TMD-N735D | *Doumazane et al., 2011* (modified) | | Following SNAP tag, mGluR2 truncatd to position Q558. N at position 735 replaced with D |
| Transfected construct (*H. sapiens*) | SNAP-B2AR | modified from addgene plasmid pSNAPf-ADRβ2 | Addgene Plasmid #101123 | |
| Transfected construct (*H. sapiens*) | GIRK1-F137S homotetramerization mutant | | | |

*Continued on next page*

*Continued*

| Reagent type | Designation | Source | Identifiers | Additional information |
|---|---|---|---|---|
| Transfected construct (synthetic) | tdTomato | Addgene | Addgene Plasmid #30530 | |
| Transfected construct | Gαiq3 | Concept from *Conklin et al., 1993* | | *Conklin et al., 1993* |
| Transfected construct | GCaMP6f | Addgene | Addgene Plasmid #73564 | |
| Antibody | Rabbit polyclonal to HA tag (Biotin) | Abcam | ab26228, RRID:AB_449023 | 10–20 nM (1:500 dilution) |
| Chemical compound, drug | NeutrAvidin | ThermoFisher | Cat # 31000 | |
| Chemical compound, drug | mPEG | Laysan Bio | Item# BIO-PEG-SVA-5K-100MG andMPEG-SVA-5K-1g | |
| Chemical compound, drug | biotinylated mPEG | Laysan Bio | Item# BIO-PEG-SVA-5K-100MG and MPEG-SVA-5K-1g | |
| Chemical compound, drug | benzylguanine (BG)-LD555 | Scott Blanchard lab (synthesized in house) | | Synthesized in-house |
| Chemical compound, drug | benzylguanine (BG)-LD655 | Scott Blanchard lab (synthesized in house) | | Synthesized in-house |
| Chemical compound, drug | IGEPAL | Sigma Aldrich | I8896-50 | |
| Chemical compound, drug | DDM | Sigma Aldrich | 850520P | |
| Chemical compound, drug | SNAP Surface Alexa-546 | New England Biolabs | Cat # S9132S | |
| Chemical compound, drug | CLIP Surface Alexa-547 | New England Biolabs | Cat # S9233S | |
| Chemical compound, drug | DMEM | Thermo Fisher Scientific | Cat # 11995073 | |
| Chemical compound, drug | Fetal Bovine Serum | Thermo Fisher Scientific | Cat # 10437028 | |
| Chemical compound, drug | Lipofectamine 3000 Transfection Reagent | Thermo Fisher Scientific | Cat # L3000015 | |
| Chemical compound, drug | Poly-L-lysine hydrobromide | Sigma Aldrich | P2636 | |
| Chemical compound, drug | Glutamate | Sigma Aldrich | Cat # 6106-04-3 | |
| Chemical compound, drug | Xanthurenic acid | Tocris | Cat # 4120 | |

*Continued on next page*

*Continued*

| Reagent type | Designation | Source | Identifiers | Additional information |
|---|---|---|---|---|
| Chemical compound, drug | LY487379 | Tocris | Cat # 3283 | |
| Chemical compound, drug | BINA | Tocris | Cat # 4048 | |
| Chemical compound, drug | Ro 64–5229 | Tocris | Cat # 2913 | |
| Chemical compound, drug | MNI 137 | Tocris | Cat # 4388 | |
| Chemical compound, drug | TASP 0433864 | Tocris | Cat # 5362 | |
| Chemical compound, drug | VU0360172 | Tocris | Cat # 4323 | |
| Chemical compound, drug | LY341459 | Tocris | Cat # 1209 | |
| Chemical compound, drug | 22:1 phosphotidylcholine | Avanti Polar Lipids | Cat # 850398C | |
| Chemical compound, drug | Aminonaphthalene trisulfonic acid | Thermo Fisher | Cat # A350 | |
| Chemical compound, drug | Thalium nitrate | Sigma Aldrich | Cat # 309230 | |
| Chemical compound, drug | Gramicidin A | Sigma Aldrich | Cat # 50845 | |
| Software, algorithm | LabVIEW | http://www.ni.com/en-us/shop/labview/labview-details.html | RRID:SCR_014325 | |
| Software, algorithm | ImageJ | http://imagej.nih.gov/ij/ | RRID:SCR_003070 | |
| Software, algorithm | GraphPad Prism | https://graphpad.com | RRID:SCR_002798 | |
| Software, algorithm | ClampX | https://www.moleculardevices.com/ | RRID:SCR_011323 | |
| Software, algorithm | Clampfit | https://www.moleculardevices.com/ | RRID:SCR_011323 | |
| Software, algorithm | Microsoft Excel | https://products.office.com/en-us/excel | RRID:SCR_016137 | |
| Software, algorithm | Origin | https://www.originlab.com/ | RRID:SCR_002815 | |
| Software, algorithm | Adobe Illustrator | https://www.adobe.com/ | RRID:SCR_010279 | |
| Software, algorithm | Olympus cellSens | www.olympus-lifescience.com/cellsens | RRID:SCR_016238 | |

## Cell culture, molecular cloning and gene expression

HEK 293T cells were purchased from ATCC, validated by DNA profiling (Bio-Synthesis, Inc) and tested negative for mycoplasma using a commercial kit. Cells were cultured in DMEM with 5% FBS on poly-L-lysine-coated glass coverslips. DNA plasmids were transfected into cells using Lipofectamine 3000 (Thermo Fisher). For electrophysiology experiments, cells were transfected with HA-SNAP-mGluR2 or HA-SNAP-mGluR2-TMD, GIRK1-F137S homotetramerization mutant (*Vivaudou et al., 1997*), and tdTomato (as a transfection marker) at a 7:7:1 ratio with 0.7 µg plasmid/well for receptor and channel. For calcium imaging, cells were transfected with HA-SNAP-mGluR2, GCaMP6f and a $G\alpha_{iq3}$ chimera (*Conklin et al., 1993*) at 7:5:3 ratio with 0.7 µg plasmid/well for receptor. For single-molecule pulldown (SiMPull) experiments, cells were transfected with 0.7 µg DNA of HA and SNAP-tagged receptor constructs, unless otherwise noted. For FRET experiments, cells were transfected with either SNAP-tagged constructs at 0.5–0.7 µg of DNA/well or SNAP and CLIP-tagged constructs at a ratio of 1:2 with 0.3 µg of SNAP DNA/well.

All SNAP-mGluR-TMD clones were made by modifying previously reported full-length SNAP-tagged mGluR cDNA (*Doumazane et al., 2011*). The LBD and CRD were deleted using 5' phosphorylated PCR primers to amplify the remaining sequence and re-ligate. Following the SNAP tag, mGluR1-TMD starts at V583, mGluR2-TMD (rat) starts at Q558, mGluR3-TMD (rat) starts at E567, and mGluR5-TMD (human) starts at V570. The ß$_2$AR vector was obtained from addgene (clone 101123). An HA tag was added immediately upstream of the SNAP tag using a PCR-based insertion with phosphorylated primers. The GFP-tagged $G\alpha_{oA}$ protein construct used was previously reported by *Lober et al. (2006)*. Mutations were made using standard PCR-based techniques.

## Whole-cell patch-clamp electrophysiology

HEK 293T cell electrophysiology was performed as previously described (*Vafabakhsh et al., 2015*). Whole-cell patch clamp recordings from single isolated cells were performed 24–36 hr after transfection in a high-potassium extracellular solution containing (in mM): 120 KCl, 29 NaCl, 1 MgCl$_2$, 2 CaCl$_2$ and 10 HEPES, pH 7.4. Cells were voltage clamped to −60 mV using an Axopatch 200B amplifier (Axon Instruments) and membrane currents were recorded. Glass pipettes of resistance between 3 and 8 MΩ were filled with intracellular solution containing (in mM): 140 KCl, 10 HEPES, 3 Na$_2$ATP, 0.2 Na$_2$GTP, 5 EGTA and 3 MgCl$_2$, pH 7.4. Data were acquired with a 2 kHz acquisition rate and filtered with the amplifier 4-pole Bessel filter at 1 kHz. Drugs were prepared in extracellular solution and applied using a gravity-driven perfusion system. Data were analyzed with Clampfit (Molecular Devices) and Prism (GraphPad). Data for all conditions came from at least two separate transfections. Inverse agonism was calculated based on the raw amplitude of the NAM response (I$_{NAM}$) and the raw amplitude of the glutamate response (I$_{Glu}$), where % inverse agonism = 100 x I$_{NAM}$/(I$_{NAM}$ +I$_{Glu}$).

## Calcium imaging

24–48 hr after transfection, cells were imaged at room temperature in extracellular solution containing (in mM): 135 NaCl, 5.4 KCl, 2 CaCl$_2$, 1 MgCl$_2$, 10 HEPES, pH 7.4. Experiments were conducted with continuous gravity-driven perfusion on an inverted microscope (Olympus IX73) and imaged with a 20x objective. During experiments, GCaMP6f was excited using a 488 nm laser and movies were recorded with an scMOS camera (Hamamatsu ORCA-Flash4v3.0) at 1 Hz with a 100 ms exposure time. PAM responses were quantified relative to the responses to 1 mM glutamate. Regions of interest were selected in Olympus cellSens software and represent single cells or small cell clusters (2–3 cells). Data analysis was performed in Microsoft Excel, where intensities were normalized to the baseline prior to drug application. Dose–response curves were fit using Prism (Graphpad) and data comes from at least two independent experiments for each LY48 concentration.

## Single-molecule pulldown and subunit counting

Single-molecule pulldown was performed as previously described (*Levitz et al., 2016*). Briefly, to prevent non-specific sticking of proteins, flow chambers were prepared with mPEG-passivated glass slides and coverslips doped with ~1% biotinylated mPEG, as previously described. Prior to each experiment, chambers were incubated with 0.2 mg/mL NeutrAvidin (ThermoFisher) for 2 min followed by 10 nM of a biotinylated anti-HA antibody (abcam ab26228) for 20–30 min. After each conjugation step, the chambers were washed with T50 buffer (50 mM NaCl, 10 mM Tris, pH 7.5).

24–48 hr after transfection, HEK 293T cells were labeled with 1.5 µM benzylguanine (BG)-LD555 in extracellular solution at 37°C for 45 min. BG-conjugated LD fluorophores are based on previously reported Cy3 and Cy5-based fluorophores with an incorporated cyclooctatetraene, a protective agent (*Zheng et al., 2017*). After labeling, cells were dissociated from coverslips by incubating with Ca$^{2+}$-free PBS for 20 min followed by gentle pipetting. Cells were pelleted by centrifugation at 16,000 g for 1 min and then lysed in buffer containing (in mM): 10 Tris, 150 NaCl, 1 EDTA, protease inhibitor cocktail, and 1.2% either IGEPAL (Sigma) or *n*-dodecyl-β-D-maltoside (DDM) (Sigma) at pH 8.0. Following 1 hr of lysis at 4°C, cells were centrifuged at 16,000 g for 20 min before the supernatant was collected and maintained on ice. The cell lysate was diluted in a 0.1% detergent dilution buffer to obtain sparse immobilization of labeled protein on the passivated slide. After obtaining an optimal density of immobilized protein, the flow chamber was washed with dilution buffer to remove unbound protein. For SiMPull experiments with PAMs and NAMs, drugs were added to cells during the dissociation step (after dye labeling), and maintained throughout the cell lysis and entire experiment.

Single molecules were imaged with a 100x objective on an inverted microscope (Olympus IX73) in total internal reflection fluorescence (TIRF) mode. Images were taken with an scMOS camera (Hamamatsu ORCA-Flash4v3.0) at 20 Hz with 50 ms exposures. A 561 nm laser was used to excite the LD555 fluorophore. Multiple independent experiments were performed for each condition. Data analysis was performed in LabVIEW as previously described (*Ulbrich and Isacoff, 2007*). A SNAP-tag labeling efficiency of ~80% was used to calculate the proportion of dimers and monomers. Occasional co-localization of two proteins within a diffraction limited spot along with antibody and NeutrAvidin bivalency both make a small contribution to >1 step photobleaching, leading to a background % two-step bleaching level of ~5–10%. All data were produced from at least two separate transfections/protein preparations for each condition.

## Live-cell FRET measurements

24–36 hr after transfection, culture media was removed from cells and coverslips were washed with extracellular (EX) solution containing (in mM): 135 NaCl, 5.4 KCl, 2 CaCl$_2$, 1 MgCl$_2$, 10 HEPES, pH 7.4. Cells were labeled at 37°C for 45 min with either 1.5 µM benzylguanine LD555 and 1.5 µM benzylguanine LD655 (for SNAP only experiments) or 1.5 µM benzylcytosine DY-547 (NEB) and 3 µM benzylguanine LD555 (for SNAP +CLIP experiments). Fluorophores were diluted in EX solution, which was also used to wash coverslips following labeling and prior to mounting on the microscope. After labeling, cells were mounted on an inverted microscope (Olympus IX73) and imaged with a 60x objective. Donor was excited using a 561 nm laser and images were taken simultaneously in the donor and acceptor channels on separate scMOS cameras (Hamamatsu ORCA-Flash4v3.0) (see *Figure 3—figure supplement 1A*) at 0.5–1 Hz with a 100 ms exposure time. Clusters of cells were analyzed together using ImageJ and FRET was calculated as FRET=($I_{Acceptor}$)/($I_{Donor}$ +$I_{Acceptor}$) where I is fluorescence intensity. No corrections were made for bleed-through between channels or the contribution of donor-donor or acceptor-acceptor dimers as analysis was limited to relative FRET changes between drug treatments rather than absolute FRET values. A small artifact (decrease in fluorescence) in response to PAM application was observed in acceptor-only controls, but this response showed the same dose-dependence, relative amplitude and kinetics as FRET responses and were, thus, not corrected for. For individual traces, FRET was normalized to the basal FRET value observed before application of drugs. FRET changes calculated for dose–response curves were normalized to the response to saturating LY483739 applied within the same recording and dose–response curves were obtained from multiple cell clusters and averaged from at least three separate experiments. Dose–response curves were fit using Prism (Graphpad). All drugs were purchased from Tocris, prepared in EX solution, and delivered with a gravity-driven perfusion system. For kinetics measurements a pressurized perfusion system (Automate Scientific) was used to produce a flow rate of 0.5 mL/s. Kinetic analysis was performed in Microsoft Excel. The time between 10% and 90% FRET increase or decrease was determined manually. For photo-bleaching/donor-recovery experiments, images were taken in donor and acceptor channels at baseline and again followed by a 1 min exposure to 640 nm laser illumination at maximum intensity. All experiments were performed at room temperature. All data were produced from at least two separate transfections for each condition.

## Bilayer modification experiments

The gramicidin-based fluorescence assay has been described previously (*Ingólfsson and Andersen, 2010*). In brief, large unilamellar vesicles (LUVs), loaded with intravesicular ANTS were prepared from $DC_{22:1}PC$ and gramicidin (weight ratio 1000:1, corresponding to a ~ 2000:1 molar ratio) using freeze-drying, extrusion and size-exclusion chromatography; the final lipid concentration was 4–5 mM; the suspension was stored in the dark at 12.5°C for a maximum of 7 days. The size distribution was determined using dynamic light scattering using an Anton Paar Litesizer TM 500 instrument; the average diameter was 133 nm, with an average Polydispersity index of 7.6% indicating that the samples are monodisperse. Before use, the LUV-ANTS stock was diluted to 200–250 μM lipid with $NaNO_3$ buffer (140 mM $NaNO_3$, 10 mM HEPES, pH 7).

The NAMs and PAMs (dissolved in DMSO) or DMSO (as control) were added to a LUV-ANTS sample and equilibrated at 25°C in the dark for 10 min. before the mixture was loaded into a stopped-flow spectrofluorometer (Applied Photophysics SX.20, Leatherhead, UK) and mixed with either $NaNO_3$ buffer or $TlNO_3$ buffer; $Tl^+$ (thallous ion) is a gramicidin channel-permeant quencher of the ANTS fluorescence. Samples were excited at 352 nm and the fluorescence signal above 455 nm was recorded in the absence (four successive trials) or presence (nine successive trials) of the quencher. All the NAM and PAM derivatives fluoresce to varying degrees and addition of these drugs to LUVs in control experiments without gA or $Tl^+$ increased the fluorescence signal, and each signal was normalized to account for the compound's fluorescence. The instrument has a dead time of <2 ms, and the next 2–100 ms segment of each fluorescence quench trace was fitted to a stretched exponential, which is a computationally efficient way to represent a sum of exponentials with a distribution of time constants, reflecting the LUV size distribution as well the fluctuations in the number of gramicidin channels in the LUV membranes:

$$F(t) = F(\infty) + (F(0) - F(\infty)) \cdot exp\left\{-(t/\tau_0)^\beta\right\}$$

where *F(t)* denotes the fluorescence intensity at time, *t*, $\tau_0$ is a parameter with units of time, and $\beta$ ($0 < \beta \leq 1$, where $\beta = 1$ denotes a homogenous sample) provides a measure of the LUV dispersity. The quench rate (*Rate*), the rate of $Tl^+$ influx, was determined at 2 ms:

$$Rate = \frac{\beta}{\tau_0} \cdot \left(\frac{t}{\tau_0}\right)^{\beta-1}\big|_{2ms}$$

The *Rate* for each experiment represents the average quench rates of the trials with $Tl^+$. The average *Rate* was normalized to the rate in control experiments ($Rate_{cntrl}$) without modulator. The reported values are averages from three or more experiments.

## Statistics and data analysis

Data was analyzed using Clampfit (Axon Instruments), Origin (OriginLab), Prism (Graphpad) and ImageJ software. Statistical analysis was performed using Microsoft Excel and Prism. All values reported are mean ± s.e.m.

## Acknowledgements

We thank Anant Menon, Javier Gonzalez-Maeso and Jeremy Dittman for helpful discussion, and Konstantinos Vlachos, Roger Altman and Joon Lee for technical support. JL is supported by an R35 grant from the National Institute of General Medical Science (1 R35 GM124731) and the Rohr Family Research Scholar Award. JKT is supported by an NSF Graduate Research Fellowship. DSS is supported by a Medical Scientist Training Program grant from the National Institute of General Medical Sciences of the National Institutes of Health under award number T32 GM007739 to the Weill Cornell/Rockefeller/Sloan Kettering Tri-Institutional MD-PhD Program and an Abby R Mauze Medical Scientist Fellowship. OSA is supported by an R01 grant from the National Institute of General Medical Science (1 R01 GM021342).

## Additional information

### Competing interests

Scott C Blanchard: SCB holds equity interest in Lumidyne Technologies. The other authors declare that no competing interests exist.

### Funding

| Funder | Grant reference number | Author |
|---|---|---|
| National Institute of General Medical Sciences | 1R35GM124731 | Joshua Levitz |
| National Institute of General Medical Sciences | 1R01GM021342 | Olaf S Andersen |
| National Institute of General Medical Sciences | R01GM098858-07 | Scott C Blanchard |

The funders had no role in study design, data collection and interpretation, or the decision to submit the work for publication.

### Author contributions

Vanessa A Gutzeit, Data curation, Formal analysis, Validation, Investigation, Visualization, Methodology, Writing—review and editing; Jordana Thibado, Formal analysis, Investigation, Visualization, Methodology, Writing—review and editing; Daniel Starer Stor, Data curation, Formal analysis, Investigation, Methodology, Writing—review and editing; Zhou Zhou, Validation, Investigation, Methodology; Scott C Blanchard, Resources, Supervision, Validation, Methodology, Writing—review and editing; Olaf S Andersen, Data curation, Formal analysis, Supervision, Validation, Methodology, Writing—review and editing; Joshua Levitz, Conceptualization, Resources, Data curation, Formal analysis, Supervision, Funding acquisition, Validation, Investigation, Visualization, Methodology, Writing—original draft, Project administration, Writing—review and editing

### Author ORCIDs

Vanessa A Gutzeit (iD) https://orcid.org/0000-0002-7714-7268
Scott C Blanchard (iD) http://orcid.org/0000-0003-2717-9365
Joshua Levitz (iD) https://orcid.org/0000-0002-8169-6323

### Decision letter and Author response

Decision letter https://doi.org/10.7554/eLife.45116.028
Author response https://doi.org/10.7554/eLife.45116.029

## Additional files

### Supplementary files

• Transparent reporting form
DOI: https://doi.org/10.7554/eLife.45116.026

### Data availability

All data generated or analyzed during this study are included in the manuscript and supporting files.

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
