## [Decision Letter]

Thank you for submitting your article "Conformational dynamics between transmembrane domains and allosteric modulation of a metabotropic glutamate receptor" for consideration by *eLife*. Your article has been reviewed Richard Aldrich as the Senior Editor, Kenton Swartz as the Reviewing Editor, and two reviewers. The reviewers have opted to remain anonymous.

The reviewers have discussed the reviews with one another and the Reviewing Editor has drafted this decision to help you prepare a revised submission.

Summary:

This paper uses a combination of cleverly designed probes inserted into isolated receptor domains and full-length mGluR2 for live cell FRET experiments and single molecule photobleaching. With these the authors investigate the mechanisms by which a battery of drugs modulate mGluR2 activity, focusing on movements in the extracellular ligand binding domains, and the transmembrane domain of the mGluR2 dimer assembly. This is a field in which work by multiple groups published over the past ten years had convincingly established that mGluRs, like other class C GPCRs, assemble as dimers, that allosteric modulators bind in a variety of sites to regulate receptor function, and that isolated mGluR2 transmembrane domains can be activated by allosteric modulators in the absence of the ligand binding domain. Most recently, cryo-EM and crystal structures for mGluR5 reveal in detail how the TMD domains interact in an intact dimer assembly. In the present study, the photo bleaching, FRET measurements, and electrophysiological data overlap with previously published work, but do add further insights into mGluR activation and modulation. The kinetics measurements of drug action are not convincing, and some of the quantitative analysis of the FRET data needs closer inspection. In the Introduction (and Abstract) the authors conspicuously ignore recent advances which answer some of the same questions their experiments address, and make some unwarranted statements, giving the false impression that less is known about mGluR assembly and function than is the case. We request that you address the following concerns, some of which will likely require reinterpretation, removing data from the manuscript and considerably rewriting.

Essential revisions:

1) The half maximum dose in the dose response is one order of magnitude different in the FRET (Figure 3F) than in the current dose response (Figure 1F), as if the two phenomena are not strongly related. What does this indicate?

2) The data in Figure 5B and Supplemental Figure 13, Figure 6—figure supplement 3C and Figure 7—figure supplement 1 are for the most part not convincing and it is hard to make mechanistic interpretations with respect to receptor modulation and gating. Although the authors measure a 10-90% rise time, the shape of individual responses differs substantially, and some of the data make no sense. What do the kinetics actually represent? Does mGluR2 activation really take minutes to reach equilibrium? Why are the off times concentration dependent for a soluble ligand like glutamate? The responses to glutamate are much slower than recorded at synapses. In Figure 6 –figure supplement 1 the slope of the concentration response to CBiPES is nearly flat compared to other ligands. The authors also find it hard to washout BINA and suggest that BINA is sequestered into the membrane. This is also probably true for some of the other compounds used (or there must be some other factor that make washout slow and concentration dependent), since the Off rate is concentration dependent (which it shouldn't be if there is a simple washout of the compound). The authors should address these kinetic issues or remove the data from the manuscript.

3) The following statements are misleading and should be corrected in light of recent full-length mGluR structures cited in the Discussion section. Abstract: "it remains unknown how mGluR activation proceeds at the level of the transmembrane domains (TMDs)". Introduction: "It remains unclear if TMDs directly interact with each other".

4) The authors propose that Ro-64 works as a reverse agonist, since it deceases the baseline current. However, isn't this just antagonism of a basal activity of the mGLUR2? MNI-137 also decreases the baseline current, but the authors propose that MNI-137 works as a neutral antagonist. Please clarify?

5) The fact that Ro-64 (a NAM) induce the same TMD FRET as LY-48 (a PAM) is of somewhat concern: doesn't this mean that the TMD FRET signal is not really a good reporter of activation?

6) The FRET equation (subsection “Live Cell FRET Measurements”) is not correct for macroscopic FRET analysis. The Fdonor is 50% from dimers with two donor fluorophores. In addition, the Facceptor is a mixture of donor bleed through, direct excitation of acceptors, and FRET excitation of acceptors. Please make corrections to FRET efficiency calculations or, at least, make it clear what the shortcomings of the present calculations are.

7) In Figure 4C-D: is the DCG-IV concentration saturating? In Figure 6D: what happens if the MNI-137 concentration is increased?

8) Subsection “PAM-induced inter-TMD FRET responses are tightly correlated with functional PAM affinity and Efficacy”. "This result suggests that the accessibilities of orthosteric and allosteric binding sites differ and that the timing of onset of downstream effects induced by agonists versus PAMs would be different." It is not clear you can distinguish accessibility from slow induced conformational changes.

9) Subsection “PAM-induced inter-TMD FRET responses are tightly correlated with functional PAM affinity and Efficacy”. "The presence of G protein accelerated both ON and OFF kinetics". Off kinetics is not different at low concentrations of LY48, so maybe no effect if Off concentration dependence is an artifact (see above point 2)?

10) Subsection “Inter-TMD FRET reveals that mGluR2 NAMs serve as either neutral antagonists or inverse agonists which stabilize a high FRET state”. "suggesting that the resting TMD conformation is not altered by this compound." You only looked at one place in the protein, so many changes could occur at other places that you haven't looked at.

11) Subsection “The role of inter-TMD interaction in mGluR modulation and activation”." indicates that these conformational changes likely represent a similar activation process to that initiated by glutamate." This seems like a stretch too, especially since the NAM induce the same FRET signal.

12) Introduction: "mGluR2.…inter-TMD reorientation is underscored by a unique, high propensity for mGluR2 TMD dimerization that is not seen in other group I and II mGluR subtypes or in canonical class A GPCRs". No question that class A GPCRs are different, but given that class C GPCRs are functional dimers, the "unique, high propensity" for mGluR2 compared to other mGluRs is doubtful, and it’s not convincingly demonstrated that the differences observed are mechanistically significant.

13) Subsection “Positive allosteric modulators directly activate mGluR2 with minimal contribution from the extracellular domains”: "Application of LY48 to cells expressing mGluR2 and GIRK produced large, reversible inward currents that were similar in amplitude to those induced by saturating glutamate (Figure 1B, Figure 1—figure supplement 1A)." This is not true; the data show the response is 70% to that of Glutamate.

14) Figure 6C: the fit to the data for MNI 137 is meaningless; both should be omitted, showing just Ro 64.

15) Figure 6 panels D-F: it’s unclear how inverse agonism in F was estimated, and how this relates to the raw data in D and E.

[Editors' note: further revisions were requested prior to acceptance, as described below.]

Thank you for resubmitting your work entitled "Conformational dynamics between transmembrane domains and allosteric modulation of a metabotropic glutamate receptor" for further consideration at *eLife*. Your revised article has been favorably evaluated by Richard Aldrich (Senior Editor), Kenton Swartz (Reviewing Editor), and two reviewers.

The reviewers all feel that the manuscript has been greatly improved but there are some remaining issues that need to be addressed before acceptance, as outlined below:

Essential revisions:

In this revised manuscript the authors do a much better job of presenting their work in the context of prior studies on mGluRs, explaining clearly how the combination of independent FRET measurements on both the LBD and TMD, in functionally active GPCRs under more controlled conditions, gives greater insight into the action of allosteric modulators and mGluR dimeric assemblies. A limitation of the study is that both positive and some negative allosteric modulators give the same FRET change in the TMD, which the authors model as reflecting two different but unknown conformations. Despite this limitation, the work makes advances that should be of interest to those working on type C GPCRs and perhaps a wider audience. While the manuscript is substantially improved, especially in the Abstract and Introduction, the revised Discussion section reads like a free association of highly speculative ideas, several of which go way beyond what the data can actually address. We encourage the authors to shorten the Discussion, especially the speculation on evolution, kinetics and studies in vivo.

---

## [Author Response]

Summary:This paper uses a combination of cleverly designed probes inserted into isolated receptor domains and full-length mGluR2 for live cell FRET experiments and single molecule photobleaching. […] We request that you address the following concerns, some of which will likely require reinterpretation, removing data from the manuscript and considerably rewriting.

We thank the referees for their constructive comments on the manuscript. We have responded to each of the specific revision requests below, which included modifications to the manuscript as well as new experiments. We have taken the reviewers’ comments to heart and made an effort to more thoroughly place our work in the context of the broader mGluR field with revisions to the Abstract, Introduction and Discussion section. We have provided responses to specific points below.

Overall, we think that our study provides a valuable contribution to two major research areas. First, we provide new insight into the mechanisms of mGluR transmembrane domain (TMD) assembly, conformation and function. Understanding the TMD as an autonomous unit is a key step toward gaining a complete view of class C GPCR activation. Second, we report findings regarding the molecular pharmacology of mGluR allosteric modulators that reveal new forms of heterogeneity between related compounds. We’d like to emphasize the following aspects of our study that either represent new findings or important clarifications to unresolved questions:

1) full-length mGluR2 can be activated by PAMs.

2) PAMs can activate mGluR2 without the need for orthosteric agonist binding or LBD closure.

3) mGluR TMDs interact with sufficient affinity to dimerize in the absence of LBDs.

4) mGluR2 TMDs show a higher propensity for dimerization than all other group I and II mGluRs.

5) inter-TMD rearrangement can occur independently of allosteric input from the LBDs.

6) PAMs have both variable affinity and variable efficacy relative to each other.

7) PAMs show slower onset and reversal of effects than glutamate.

8) PAMs have variable reversibility and OFF kinetics due, in part, to lipid bilayer interactions.

9) Xanthurenic acid, a proposed endogenous allosteric agonist, does not work directly on mGluR2 TMDs.

10) NAMs may be neutral or serve as inverse agonists.

11) At least 3 states are required to describe inter-TMD conformational dynamics.

Essential revisions:1) The half maximum dose in the dose response is one order of magnitude different in the FRET (Figure 3F) than in the current dose response (Fig1F), as if the two phenomena are not strongly related. What does this indicate?

We thank the referees for pointing out the lack of correspondence between the EC_50_ values observed in the TMD FRET assay and the GIRK current assay. It’s not unusual for functional and conformational or binding readouts of GPCRs, or any signaling protein, to differ because many factors contribute to the dose response curve shape and midpoint. For example, receptor reserve pools may contribute to a leftward shift of the EC_50_ in a functional assay relative to the K_d_ measured in binding assays (for examples see Adler et al., 1987 or Adham et al., 1993). In addition, non-linearity/amplification in the downstream signaling steps may also contribute to such discrepancies between binding, conformational and functional dose-response curves. For example, positive cooperativity in the relationship between G_ß_ binding and GIRK current (Berlin et al., 2010; Yakubovich et al., 2015) likely contributes to a leftward-shift in the dose-response. Along these lines, it’s worth noting that our prior measurements of inter-LBD FRET also show a quantitatively similar discrepancy between the EC_50_ measured with FRET versus GIRK current measurements. In Levitz et al., (2016), we reported an EC_50_ value for glutamate of ~20 µM and ~3 µM in inter-LBD FRET and GIRK function, respectively. This seven-fold ratio is in line with six-fold ratio reported here for inter-TMD FRET (EC_50_ ~ 1.2 µM) and GIRK function (EC_50_ ~ 0.2 µM).

To further probe the response to PAMs, we have performed new experiments with another functional readout. Figure 1—figure supplement 1c-d shows a representative trace and an LY48 dose response curve produced from calcium imaging in HEK 293T cells, another commonly used GPCR readout. A widely used G protein chimera (Conklin et al., 1993) was co-transfected to allow mGluR2 to signal via the Gq pathway. In this system, we measured an EC_50_ of 1.2 µM, which is the same as measured with the TMD FRET readout, but right-shifted relative to GIRK. This confirms the agonism of PAMs reported in our study and further supports the notion that different GPCR readouts show subtle shifts in dose response curves.

Finally, it is also worth noting that the relative apparent affinity values reported here between different PAMs are consistent between assays and consistent with published values (Johnson et al., 2003; Bonnefous et al., 2005; Johnson et al., 2005; Hiyoshi et al., 2014). Furthermore, the relative efficacies of the three PAMs tested are in agreement between FRET and GIRK current assays (Figure 4H, I). Together, these observations support the connection between the conformational changes detected by the FRET signal and receptor activation.

2) The data in Figure 5B and Supplemental Figure 13, Figure 6—figure supplement 3 and Figure 7—figure supplement 1are for the most part not convincing and it is hard to make mechanistic interpretations with respect to receptor modulation and gating. Although the authors measure a 10-90% rise time, the shape of individual responses differs substantially, and some of the data make no sense. What do the kinetics actually represent? Does mGluR2 activation really take minutes to reach equilibrium? Why are the off times concentration dependent for a soluble ligand like glutamate? The responses to glutamate are much slower than recorded at synapses. In Figure 6 –figure supplement 1 the slope of the concentration response to CBiPES is nearly flat compared to other ligands. The authors also find it hard to washout BINA and suggest that BINA is sequestered into the membrane. This is also probably true for some of the other compounds used (or there must be some other factor that make washout slow and concentration dependent), since the Off rate is concentration dependent (which it shouldn't be if there is a simple washout of the compound). The authors should address these kinetic issues or remove the data from the manuscript.

We thank the referees for expressing their concerns about the kinetics data, which we have now improved with both more experiments and further analyses. The primary conclusions we would like to emphasize in this section are: (1) PAM responses exhibit slower ON and OFF kinetics than glutamate responses and (2) that BINA shows much slower ON kinetics and a lack of reversibility compared to other PAMs due, at least in part, to bilayer interactions.

Upon further inspection of our perfusion conditions, we decided to repeat these experiments with faster perfusion (25 mL/min versus 5 mL/min in a 0.5 mL chamber). Assuming a well-stirred chamber, the time constants for the perfusion is ~1.2 s. In the case of glutamate-induced LBD FRET responses, we believe that the kinetics likely are merely a reflection of drug wash-in and wash-out. As the referees pointed out, mGluRs should be poised to signal even more rapidly in response to glutamate than the ~1 s wash-in time observed in our measurements (see Reiner and Levitz, 2018). However, the significantly slower ON kinetics of LY48, which are maintained with fast perfusion, show that either binding or binding-induced conformational changes are indeed slower for TMD-targeting PAMs. Consistent with this interpretation, full-length mGluR2 responses in the GIRK assay are faster for saturating (1 mM) glutamate than for saturating (10 µM) LY48 and LY48-induced changes in LBD FRET are also much slower than glutamate-induced FRET changes (see new Figure 5—figure supplement 1C-D). To emphasize this comparison, we have now broken up the kinetics data into two main figures: Figure 5 shows a comparison between inter-LBD glutamate-induced kinetics and inter-TMD LY48-induced kinetics and Figure 6 shows a comparative analysis of different PAMs. Notably, the speed of different PAMs is only slightly enhanced with fast perfusion (see Figure 6A,B) and all PAMs, despite showing differences relative to each other, show slower kinetics than glutamate. In Figure 5—figure supplement 1A and B we show comparisons between a calculated time course for drug exchange into the bath, the expected binding curve for a concentration ten times the EC_50_, and the raw FRET responses for 100 µM glutamate with LBD FRET (A) or for 10 µM LY48 with TMD FRET (B).

Along the lines of this analysis, an important point we have added to the Discussion section is that the lack of a binding site for endogenous ligands within the TMD, suggests that there has been limited evolutionary constraints on this binding site. This, as previously suggested in many pharmacological contexts (see Niswender and Conn, 2010), could explain why subtype-specific ligands are easier to develop for this site, and also is consistent with the slow and variable kinetics observed.

A secondary observation is that PAM (LY48, CBiPES and TASP) OFF kinetics, but not glutamate OFF kinetics, are weakly dose-dependent (see Figure 5C and Figure 6—figure supplement 1A). Given our rapid perfusion conditions, this dose-dependence is unlikely to be due simply to re-binding as a consequence of slow wash-out. Slow wash-out from lipidic compartments, which has previously been shown to produce dose-dependent OFF kinetics for hydrophobic drugs targeting GABAA receptors (see Gingrich et al., 2009), is the likely explanation for this result, which we discuss in subsection “Conformational and Functional Diversity of mGluR Allosteric Modulators**”**. An extreme case of such an interaction with the membrane is seen with BINA which is very hydrophobic (cLogP ≈ 7.8). In contrast to the 3 other PAMs tested, BINA responses in both the TMD FRET (Figure 6A) and GIRK assays (Figure 6C) were completely irreversible, even under high speed perfusion. We believe this is an important result that can be explained by a membrane partitioning model. In figure 6D, we show that the NAM MNI 137 reverses the BINA-induced FRET increase, but that a BINA response returns following MNI washout. This result points to a population of BINA molecules that are unable to be washed out because they are partitioned either into the plasma membrane or intracellular membranes. In addition, we use a well-established in vitro liposome-based assay (Ingolfsson and Andersen, 2010) to show that BINA, to a greater extent than all other PAMs, alters the physical properties of the membrane. To clarify this measurement, we have provided a schematic and representative traces to more clearly describe this experiment (Figure 6—figure supplement 3A,B). We have also added information on reported logP values for each PAM and NAM tested to the text to provide more context.

To further probe PAM kinetics, we have also reported the effects of a dominant negative G protein, which stabilizes the active state of the TMDs (Figure 6—figure supplement 2). As reported in the original manuscript, G protein accelerated both the ON and OFF kinetics of the response to LY48. We now include data obtained with rapid perfusion where the general acceleration effect is maintained and dose-dependence of the OFF kinetic response is clear. The effect observed for ON kinetics is consistent with the enhanced apparent affinity that G protein enables. The counterintuitive effect on the OFF kinetics suggests, as we posited in the initial submission, that G protein stabilizes an active state with enhanced accessibility with the extracellular solution to enable more rapid reversal.

Finally, we also report updated kinetic data for Ro 64 obtained with fast perfusion. Consistent with the data in the original manuscript, Ro 64 shows ON kinetics that are comparable to LY48 (Figure 7—figure supplement 1B) but OFF kinetics that are much slower than those of LY48, TASP or CBiPES. Given the lack of clear effect on the membrane in our in vitro assay (Figure 7—figure supplement 1C) we suggest that this indicates that, in contrast to the G protein effect on the active state, the fully inactive state shows decreased ligand accessibility. As mentioned in the Discussion section, this is in line with recent work showing that the accessibility of GPCR ligands may be state-dependent (Devree et al., 2016). However, we also note in the discussion that the hydrophobic nature of Ro 64 (cLogP ≈ 4.9) may also contribute to its slow reversal.

The aforementioned experimental and analytical modifications may be found highlighted throughout the Results section and Materials and methods section.

3) The following statements are misleading and should be corrected in light of recent full-length mGluR structures cited in the Discussion section. Abstract: "it remains unknown how mGluR activation proceeds at the level of the transmembrane domains (TMDs)". Introduction: "It remains unclear if TMDs directly interact with each other".

We apologize if these statements were perceived as dismissive of the progress made throughout the mGluR field on the crucial question of inter-TMD interaction and receptor activation; that was certainly not our intention. At the time of manuscript preparation and initial submission, the full-length mGluR5 structure was not yet reported.

Nonetheless, we maintain that the general message of our initial statements is valid. In brief, the general view of the field has been that mGluR dimerization is mediated primarily by inter-LBD interactions based on the presence of LBD dimer crystal structures for group I, II and III subtypes (Kunishima et al., 2010; Tsuchiya et al., 2002; Muto et al., 2007; Monn et al., 2015; Koehl et al., 2019), trFRET measurements of receptor constructs lacking the LBD (El Moustaine et al., 2012) and indications that GPCR transmembrane domains do not typically form stable dimers (see Sleno and Hebert, 2018). However, the relatively well-understood inter-LBD rearrangements that drive activation (Olofsson et al., 2014; Vafabakhsh et al., 2015; Levitz et al., 2016) have raised the question of inter-TMD interaction and rearrangement. A number of elegant FRET studies revealed inter-TMD rearrangement in full-length receptors following agonist binding based on the insertion of fluorescent proteins into intracellular loops or at the C-terminus (Tateyama et al., 2004; Tateyama and Kubo, 2006; Marcaggi et al., 2009; Hlavackova et al., 2012). These studies, which focused on group I mGluRs, only assessed the effects of orthosteric agonists, and were performed in constructs that were unable to couple to G proteins raising the need for further work in this area. At the same time, atomic resolution structures have provided further motivation for the study of inter-TMD interactions. More recently, a NAM-bound crystal structure of the mGluR1 TMD showed cholesterol-mediated dimerization (Wu et al., 2014) but NAM-bound structures of the mGluR5 TMD are monomeric (Dore et al., 2014; Christopher et al., 2015). Unfortunately, no structural information is available regarding group II or III transmembrane domains. However, a landmark study by Xue et al., (2015) showed inter-TMD cross-linking in both inactive and active receptor states. Crucially, all prior studies were not able to distinguish the relative contribution of LBD-driven TMD rearrangements versus those that are driven autonomously at the level of the TMDs, particularly in response to PAMs and NAMs. Most recently, full-length cryo-EM structures of mGluR5 further motivated the need to understand the nature of inter-TMD interactions and the mechanism of action of allosteric drugs. Koehl et al., (2019) found clear inter-TMD interaction between mGluR5 subunits via TM6 in a structure of a glutamate, nanobody and PAM-bound receptor in detergent micelles. Crucially, the TMDs remain in an inactive state in this structure. Interestingly, in the apo structure no clear interaction between TMDs is observed although TM4 and TM5 are positioned opposite each other. It is worth noting that, in contrast to the agonist-bound structure, this structure was solved in lipid nanodiscs rather than detergent. While undeniably a groundbreaking study, this work doesn’t clearly decipher the modes of interaction and rearrangement of TMDs during the activation process in a biological membrane. Furthermore, it is highly likely that different receptor subtypes show differences in intra- and inter-subunit interactions that allow them to be tuned to their distinct physiological contexts in different synaptic localizations that sense different glutamate dynamics.

Together, we believe that this valuable body of work identified the need for further investigations into the ability of mGluR TMDs to interact in different functional states, their ability to rearrange independently of allosteric drive from the LBDs and the functional and conformational effects of allosteric drugs. We chose to focus on mGluR2 because it is the best understood subtype in terms of the initial conformational changes that occur at the LBD and because of the neurobiological and clinical motivation to understand allosteric drugs that target the TMDs of this receptor. The aforementioned open questions, and the associated literature, are explicitly mentioned in the introduction and throughout the Results section and Discussion section to allow the reader to place our study in the broader context of the field.

To clarify the language and more explicitly acknowledge the insight gained from the recent structures, we have modified these specific statements to read that “it remains unclear how mGluR activation proceeds at the level of the transmembrane domains (TMDs)” and “it remains unclear if TMDs can form stable interactions, whether any such interactions are state-dependent and if potential inter-TMD rearrangements are driven autonomously or depend on allosteric input from the LBDs.”

We initially aimed to keep the Introduction succinct and simple to avoid an unnecessarily detailed discussion and opted to include a more complete description of the relevant prior studies as they pertained to specific experiments throughout the Results section. However, in the revised version we have included more discussion of prior work, as well as the most recent structural study, in the Introduction.

4) The authors propose that Ro-64 works as a reverse agonist, since it deceases the baseline current. However, isn't this just antagonism of a basal activity of the mGLUR2? MNI-137 also decreases the baseline current, but the authors propose that MNI-137 works as a neutral antagonist. Please clarify?

We thank the referees for pointing out the need to clarify this point. By definition an inverse agonist decreases basal activation while a neutral antagonist merely competes for a binding site to inhibit the effects of other drugs (see Wacker et al., 2017 for a recent, comprehensive review of GPCR pharmacology). Mechanistically, this is thought to be a result of the relative affinity of the ligand for different receptor conformations (see Manglik et al., 2015; Staus et al., 2016). We propose that Ro 64 is an inverse agonist because it both shifts the conformation of the TMDs in the absence of any other drug (Figure 6B,C) and because it decreases basal activity of full-length mGluR2 (Figure 6E,F). In contrast, MNI-137 has no detectable effect on receptor conformation in our assay (Figure 6B,C) and also does not produce a substantial change in the basal current evoked by mGluR2 (Figure 6D,F).

5) The fact that Ro-64 (a NAM) induce the same TMD FRET as LY-48 (a PAM) is of somewhat concern: doesn't this mean that the TMD FRET signal is not really a good reporter of activation?

We agree that this is a counter-intuitive result. Indeed, we initially expected NAMs to either have no direct effect on TMD FRET (as was seen with MNI 137) or to decrease FRET based on the increase in FRET observed with PAMs. Such a result would have supported a 2-state model of inter-TMD conformation. We agree that the increase in TMD FRET produced by Ro 64 lends itself to some complications if one were to use this assay naively to screen drugs. However, we also find that the conformational complexity that this FRET increase revealed is an important, unexpected aspect of this paper. We finally note that, because Ro 64 has minimal bilayer-modifying effects (Figure 7—figure supplement 1C), we can exclude that the Ro 64-induced changes in FRET are due to a non-specific membrane-mediated TMD rearrangement.

6) The FRET equation (subsection “Live Cell FRET Measurements”) is not correct for macroscopic FRET analysis. The Fdonor is 50% from dimers with two donor fluorophores. In addition, the Facceptor is a mixture of donor bleed through, direct excitation of acceptors, and FRET excitation of acceptors. Please make corrections to FRET efficiency calculations or, at least, make it clear what the shortcomings of the present calculations are.

We agree with the referees about the complexity of calculating reliable FRET values from ensemble measurements in live cells. For this reason, we did not report, or discuss, actual, corrected FRET values or the associated inter-fluorophore distances, but have focused instead on relative FRET changes in response to drug application. Because of the emphasis on relative FRET changes, we did not correct the FRET values. Instead, we have defined the basal FRET level (as determined by the equation efficiency=F_A_/[F_D_ +F_A_]) as 1.0 and only analyzed relative FRET changes. Furthermore, all analysis of TMD FRET amplitude has been relative to saturating LY48. To further clarify this point, we have added a description of this to the Materials and methods section.

7) In Figure 4C-D: is the DCG-IV concentration saturating? In Figure 6D: what happens if the MNI-137 concentration is increased?

Yes, 10 µM is either saturating or very nearly saturating (~EC_80_-EC_90_) for DCG-IV in the LBD FRET assay as previously reported (Doumazane et al., 2013; Vafabakhsh et al., 2015). We chose this concentration because it has previously been shown by Doumazane et al., that the primary effect of PAM application is to increase the maximum inter-LBD FRET response, rather than to increase the apparent agonist affinity.

Similarly, 1 µM is a saturating concentration for MNI-137. Initial characterization (Hemstapat et al., 2007) across a range of assays has shown an IC_50_ in the 10-70 nM range. In addition, we found that 1 µM fully blocked the functional response to LY48 (Figure 1—figure supplement 1B). For comparison, Ro 64 has been reported to have a higher IC_50_ of 100-500 nM (Cartmell et al., 1998; Kalczewski et al., 1999) which is why we used 10 µM as a saturating dose. However, to confirm this result we have repeated the functional measurements of inverse agonism with 5 µM MNI-137 and observed identical results. Author response image 1 summarizes this new data set with 5 µM MNI-137 and 10 µM Ro 64. We chose to keep the original data in the manuscript as the concentrations used match LBD FRET data.

8) Subsection “PAM-induced inter-TMD FRET responses are tightly correlated with functional PAM affinity and Efficacy”. "This result suggests that the accessibilities of orthosteric and allosteric binding sites differ and that the timing of onset of downstream effects induced by agonists versus PAMs would be different." It is not clear you can distinguish accessibility from slow induced conformational changes.

We agree and have amended the sentence to include the possibility that differences in the timing of conformational rearrangements may also explain the discrepancy in kinetics observed for orthosteric versus allosteric drugs. Our revised sentence reads, “This result suggests that either the accessibilities of orthosteric and allosteric binding sites or the timing of the associated conformational changes differ and that the timing of onset of downstream effects induced by agonists versus PAMs would be different.”

9) Subsection “PAM-induced inter-TMD FRET responses are tightly correlated with functional PAM affinity and Efficacy”. "The presence of G protein accelerated both ON and OFF kinetics". Off kinetics is not different at low concentrations of LY48, so maybe no effect if Off concentration dependence is an artifact (see above point 2)?

We thank the referees for suggesting this. As described above (see point 2), we have repeated this experiment with faster perfusion and observed that G protein acceleration of both ON and OFF kinetics is maintained across all concentrations (see Figure 6—figure supplement 2).

10) Subsection “Inter-TMD FRET reveals that mGluR2 NAMs serve as either neutral antagonists or inverse agonists which stabilize a high FRET state”. "suggesting that the resting TMD conformation is not altered by this compound." You only looked at one place in the protein, so many changes could occur at other places that you haven't looked at.

We agree with the referee’s skepticism about the ability to draw such a conclusion from FRET measurements with a single fluorophore position. However, we wanted to highlight that of all of the drugs tested, this is the only one that did not produce a FRET change on its own. This is consistent with a conformationally neutral response, but we can’t rule out the possibility that a conformational change that our sensors are not sensitive to does actually occur. We have altered the sentence to read, “…suggesting that the TMD conformation is not altered, although we recognize that our sensors may not be able to detect more localized conformational changes”.

11) Subsection “The role of inter-TMD interaction in mGluR modulation and activation”." indicates that these conformational changes likely represent a similar activation process to that initiated by glutamate." This seems like a stretch too, especially since the NAM induce the same FRET signal.

To avoid reaching beyond what the data shows, we have modified it to read: “indicates that these conformational changes may represent a related activation pathway to that initiated by glutamate”.

12) Introduction: "mGluR2.…inter-TMD reorientation is underscored by a unique, high propensity for mGluR2 TMD dimerization that is not seen in other group I and II mGluR subtypes or in canonical class A GPCRs". No question that class A GPCRs are different, but given that class C GPCRs are functional dimers, the "unique, high propensity" for mGluR2 compared to other mGluRs is doubtful, and it’s not convincingly demonstrated that the differences observed are mechanistically significant.

Thank you. To clarify, when we discuss the unique dimerization propensities of mGluR2 we are referring strictly to the inter-TMD interaction. As previously reported by many groups, all full-length mGluRs form constitutive dimers to comparable degrees (Doumazane et al., 2011; Levitz et al., 2016), due in part to the conserved inter-subunit disulfide bond that likely renders the assembly irreversible. However, whether the TMDs directly interact with each other at rest and if this changes during activation has been controversial.

Here we find that the isolated TMDs of all mGluR subtypes tested show a propensity to dimerize above background levels, as defined by comparison to measurements with β-2 adrenergic receptors. However, the mGluR2-TMD showed a significantly higher fraction of 2-step photobleaching compared to the mGluR5-TMD and the mGluR3-TMD. This suggests that the nature of the inter-TMD interactions is different among mGluR subtypes. Though we don’t anticipate that these differences alter the stoichiometry of full-length receptors, the relative strength of the interactions between TMDs may tune the activation properties for different dimer combinations. Furthermore, the increased TMD dimerization strength of mGluR2 may explain why inactive state cross-linking of mGluR2 was observed in Xue et al., (2015), but that TMD interface was only observed in an agonist-bound state in the mGluR5 cryo-EM study (Koehl et al., 2019).

To extend this analysis, we have performed the same experiment on the isolated, SNAP-tagged TMD of mGluR1 and observed ~30% 2-step photobleaching which was similar to the TMD of mGluR3 and mGluR5 but significantly less than mGluR2 and more than the β-2-adrenergic receptor (see updated Figure 2D). This result further supports the general dimerization propensity of mGluR TMDs and the uniqueness of the mGluR2 TMD dimer interface.

13) Subsection “Positive allosteric modulators directly activate mGluR2 with minimal contribution from the extracellular domains”: "Application of LY48 to cells expressing mGluR2 and GIRK produced large, reversible inward currents that were similar in amplitude to those induced by saturating glutamate (Figure 1B, Figure 1—figure supplement 1A)." This is not true; the data show the response is 70% to that of Glutamate.

We apologize for this inaccuracy and have modified the text accordingly.

14) Figure 6C: the fit to the data for MNI 137 is meaningless; both should be omitted, showing just Ro 64.

We thank the referees for pointing out this error. We did not mean to imply that a fit was possible to the MNI-137 data. We have chosen to keep the data points without any fit in the plot for comparison to Ro 64 as we would like to make it clear that Ro 64, but not MNI-137, produces a measurable FRET change.

15) Figure 6 panels D-F: it’s unclear how inverse agonism in F was estimated, and how this relates to the raw data in D and E.

We thank the referees for pointing out the need to clearly explain how inverse agonism was calculated. We have added a description of this to the Materials and methods section. In brief, inverse agonism was calculated as the absolute value of the amplitude of the response to NAM divided by the total current. The total current was defined as the sum of the amplitude of the NAM response plus the amplitude of the glutamate response.

[Editors' note: further revisions were requested prior to acceptance, as described below.]

Essential revisions:

*In this revised manuscript the authors do a much better job of presenting their work in the context of prior studies on mGluRs, explaining clearly how the combination of independent FRET measurements on both the LBD and TMD, in functionally active GPCRs under more controlled conditions, gives greater insight into the action of allosteric modulators and mGluR dimeric assemblies. A limitation of the study is that both positive and some negative allosteric modulators give the same FRET change in the TMD, which the authors model as reflecting two different but unknown conformations. Despite this limitation, the work makes advances that should be of interest to those working on type C GPCRs and perhaps a wider audience. While the manuscript is substantially improved, especially in the Abstract and Introduction, the revised Discussion section reads like a free association of highly speculative ideas, several of which go way beyond what the data can actually address. We encourage the authors to shorten the Discussion, especially the speculation on evolution, kinetics and studies* in vivo.

We thank the referees for their positive assessment of the revised manuscript and their constructive suggestions for further improvement. As suggested, we have streamlined the discussion to remove passages that are overly speculative and stray from the data presented in this study. The modifications are highlighted in yellow in the revised manuscript and include the following:

-We removed a sentence challenging the interpretation of previous neurophysiological and behavioral studies with PAMs as our data does not directly address this point (Discussion section).

-We have removed discussion of potential differences in the allosteric binding site between ligands. Given the lack of reliable structural data in this area, we have chosen instead to simply state that differences in allosteric efficacy are likely a result of distinct poses within a common binding site.

-We deleted the short passage about the lack of evolutionary constraint on the allosteric binding site to avoid speculation (Discussion section).

-We removed conjecture about the potential role of membrane-drug interactions in receptor activation kinetics and discussion about possible state-dependent access of NAMs to the allosteric binding site (Materials and methods section).

-We removed the conjecture about the potential differences of mGluR2 PAMs in vivo and the potential for biased allosteric ligands for mGluRs (Materials and methods section). We now simply mention that our work motivates comparative studies of the efficacy and timing of different allosteric modulators.